# Breaking Silos: Adaptive Model Fusion Unlocks Better Time Series Forecasting

**Zhining Liu** [1] **Ze Yang** [1] **Xiao Lin** [1] **Ruizhong Qiu** [1] **Tianxin Wei** [1] **Yada Zhu** [2] **Hendrik Hamann** [2,3] **Jingrui He** [1] **Hanghang Tong** [1]

## Abstract

Time-series forecasting plays a critical role in many real-world applications. Although increasingly powerful models have been developed and achieved superior results on benchmark datasets, through a fine-grained sample-level inspection, we find that (i) no single model consistently outperforms others across different test samples, but instead (ii) each model excels in specific cases. These findings prompt us to explore how to adaptively leverage the distinct strengths of various forecasting models for different samples. We introduce TIMEFUSE, a framework for collective time-series forecasting with *sample-level adaptive fusion* of heterogeneous models. TIMEFUSE utilizes meta-features to characterize input time series and trains a learnable fusor to predict optimal model fusion weights for any given input. The fusor can leverage samples from diverse datasets for joint training, allowing it to adapt to a wide variety of temporal patterns and thus generalize to new inputs, even from unseen datasets. Extensive experiments demonstrate the effectiveness of TIMEFUSE in various long-/short-term forecasting tasks, achieving near-universal improvement over the state-of-the-art individual models. Code is available at https://github.com/ZhiningLiu1998/TimeFuse.

## 1. Introduction

Time series forecasting is pivotal in a variety of real-world scenarios and has been studied with immense interest across many domains, such as finance (Sezer et al., 2020), energy management (Deb et al., 2017; Hoffmann et al., 2020), traffic planning (Li et al., 2015; Yuan & Li, 2021), healthcare (Ye et al., 2023), and climate science (Lim & Zohren,

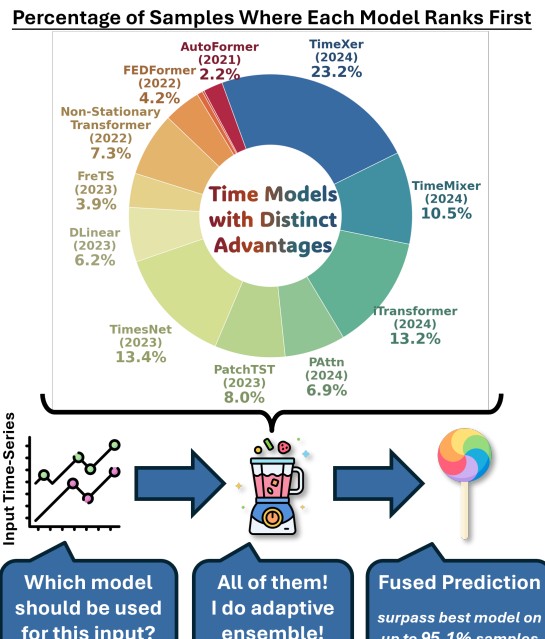

**Figure 1.** Sample-level inspection[2] reveals that each time series model excels in a considerable fraction of test samples, highlighting their unique strengths for certain types of input. TIMEFUSE adaptively leverages the strengths of different models for each input time series, achieving dynamically fused forecasting that outperforms the best individual model on up to 95.1% samples.

2021; Fu et al., 2025). Due to the complexity and dynamics of real-world systems, observed time series often exhibit intricate temporal characteristics arising from a mixture of seasonality, trends, abrupt changes, and multi-scale dependencies, which in turn present significant challenges for time-series modeling (Lim & Zohren, 2021; Wang et al., 2024a). In recent years, researchers have been striving to create increasingly sophisticated models (Zhou et al., 2022; Wu et al., 2023; Wang et al., 2024c) designed to capture and predict such temporal dynamics more effectively.

Despite the significant advancements in time series modeling with increasingly better performance achieved on benchmark datasets, a closer inspection applying a more fine-

---

[1]University of Illinois Urbana-Champaign [2]IBM Research [3]Stony Brook University. Correspondence to: Zhining Liu <liu326@illinois.edu>, Hanghang Tong <htong@illinois.edu>.

*Proceedings of the $42^{nd}$ International Conference on Machine Learning*, Vancouver, Canada. PMLR 267, 2025. Copyright 2025 by the author(s).

[2]Results collected using 14 models on 7 forecasting datasets with 96 prediction steps following the optimal settings for each model reported in Wu et al. (2023) and Wang et al. (2024b).

grained, sample-level lens presents quite a different picture, as illustrated in Figure 1. Specifically, we analyzed the performance of popular high-performing models across 7 commonly used forecasting datasets on all test samples and presented the percentage of samples where each model ranked first. Our analysis highlights two intriguing findings: **(i) there is no universal winner**: even the latest models that achieve state-of-the-art results on most benchmark datasets, were top-performing on only up to 23.2% of test samples; **(ii) each model has distinct and notable strengths**: even for the bottom-ranked models, they still ranked first on a non-negligible fraction of test samples. These findings underscore that no single model consistently outperforms the others, but instead, each model bears its unique strengths and weaknesses, often excelling in capturing specific types of temporal patterns, e.g., TimeMixer (Wang et al., 2024a) with explicit multiscale mixing is good at handling samples with high spectral complexity, while non-stationary transformer (Liu et al., 2022) can be more suitable for handling samples with low stationarity. This highlights the potential benefits of a strategy that does not rely solely on a single model, which prompts our research question:

*How can we harness different models' diverse and complementary capabilities for better time-series forecasting?*

In answering this question, we introduce **TIMEFUSE**, a novel ensemble time-series forecasting framework that enables **sample-level adaptive fusion** of heterogeneous models based on the unique temporal characteristics of each input time series. Unlike traditional ensemble strategies that statically combine different models, TIMEFUSE learns a dynamic fusion strategy that *adaptively combines models at test time*, unlocking the potential of model diversity in a highly targeted manner. Specifically, TIMEFUSE contains two core components: a meta-feature extractor that captures the temporal patterns of the input time series, and a learnable fusor that predicts the optimal combination of model outputs for each input. We employ a diverse set of meta-features to comprehensively characterize the input series, including statistical (e.g., skewness, kurtosis), temporal (e.g., stationarity, change rate), and spectral (e.g., dominant frequency, spectral entropy) descriptors. The fusor then leverages the meta-features to predict the optimal weights for combining the base models. During meta-training, we train the fusor to minimize the fused forecasting error across a broad spectrum of samples with diverse dynamics, thus improving its adaptability to unseen temporal patterns.

The design of TIMEFUSE bears several key advantages: **(i) Versatility:** The meta-training process of TIMEFUSE is decoupled from the training of base models, allowing us to integrate various models with diverse architectures into the model zoo and harness their distinct strengths for joint forecasting. **(ii) Generalizability:** Benefiting from the us-

age of meta-features, the fusor can be jointly trained using samples from different datasets, thereby generalizing to a broader range of temporal patterns and input characteristics, and thus achieving strong zero-shot performance on datasets unseen during meta-training. **(iii) Interpretability:** TIMEFUSE operates in a transparent and interpretable manner. Users can examine the fusor outputs to see how different models contribute to the fused prediction. Additionally, the learned fusor weights offer insights into how specific input temporal properties (e.g., stationarity, spectral complexity) align with the strengths of different models. **(iv) Performance:** Extensive experiments and analysis confirm TIMEFUSE's efficacy in real-world forecasting tasks. By adaptively leveraging the unique strengths of different models, TIMEFUSE unlocks more accurate predictions than the state-of-the-art base model on up to 95.1% of samples, achieving near-universal performance improvements across various benchmark long/short-term forecasting tasks.

To sum up, our contributions are in threefold: **(i) Novel Framework:** We introduce a novel framework that transitions the focus from an individual model to a *sample-level adaptive ensemble* approach for time series forecasting. This shift provides a fresh perspective and promotes a more holistic understanding of model capabilities. **(ii) Practical Algorithm:** We present TIMEFUSE, a versatile solution for fine-grained adaptive time-series model fusion. It learns and selects the optimal model combination based on the characteristics of the input time series in an interpretable and adaptive manner, thereby unlocking more accurate predictions. **(iii) Empirical Study:** Systematic experiments and analysis across a diverse range of real-world tasks and state-of-the-art models validate the effectiveness of TIMEFUSE, highlighting its strong capability as a versatile tool for tackling complex time series forecasting challenges.

## 2. Preliminaries

**Notations** We begin by defining the basic notations and concepts used in this work. For multivariate time-series forecasting with $d$ variables, the objective is to predict the values of each variable over the next $T_{\text{out}}$ time steps based on observations from the most recent $T_{\text{in}}$ time steps. Formally, let $\boldsymbol{X}_{\text{in}} \in \mathbb{R}^{T_{\text{in}} \times d}$ denote an input time series, where $T_{\text{in}}$ represents the number of input time steps, and $d$ is the number of variables or features. Formally, let $\boldsymbol{X}_{\text{in}} \in \mathbb{R}^{T_{\text{in}} \times d}$ denote an input time series, the forecast output is denoted as $\boldsymbol{X}_{\text{out}} \in \mathbb{R}^{T_{\text{out}} \times d}$. A base time forecasting model is represented as a function $f : \mathbb{R}^{T_{\text{in}} \times d} \to \mathbb{R}^{T_{\text{out}} \times d}$, mapping the input time series to the predicted output series. Many models with distinct architectural designs (Zhou et al., 2022; Wu et al., 2023; Wang et al., 2024a) have been developed to capture various temporal patterns for precise forecasting.

In this work, we utilize multiple forecasting models, collectively forming a set $\mathcal{F} = \{f_1, f_2, \ldots, f_k\}$, which we

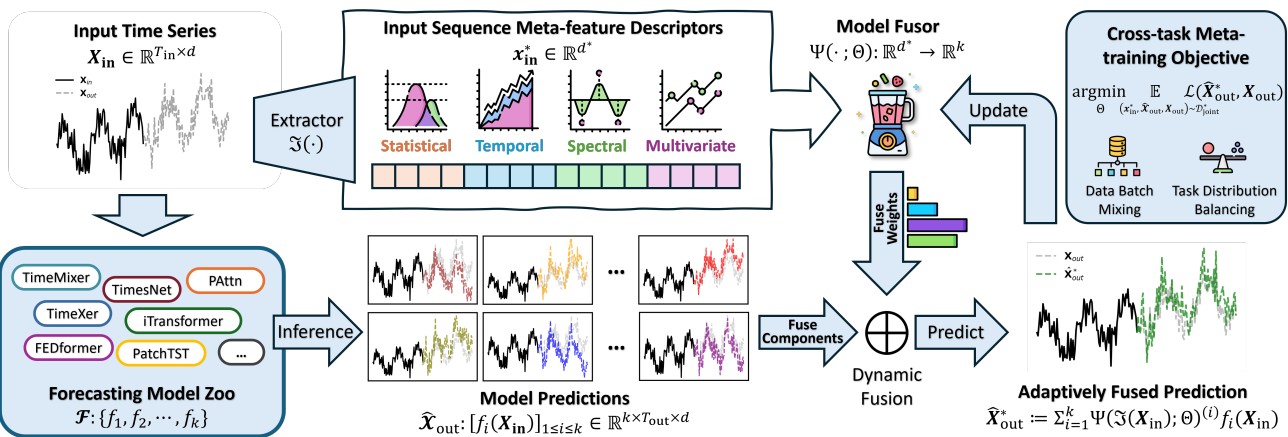

*Figure 2.* The TIMEFUSE framework for ensemble time-series forecasting, best viewed in color.

refer to as the *model zoo*. Each $f_i$ in $\mathcal{F}$ is an independently trained model, offering diverse predictions for a given input. As shown in Figure 1, different time series modeling techniques offer unique advantages; therefore, we aim to develop a fusion mechanism to adaptively harness the diverse and complementary capabilities of different models. Formally, we provide the problem definition as follows.

*Problem* 1 (Sample-level Adaptive Fusion for Time-Series Forecasting). Consider a model zoo $\mathcal{F} = \{f_1, f_2, \ldots, f_k\}$ that consists of $k$ independently trained time-series forecasting models $f_i : \mathbb{R}^{T_{in} \times d} \rightarrow \mathbb{R}^{T_{out} \times d}$ with diverse architectures. The problem is to design an adaptive ensemble forecasting mechanism that for any input time series $\boldsymbol{X}_{in}$, it dynamically leverages the strengths of each model in $\mathcal{F}$ based on the characteristics of $\boldsymbol{X}_{in}$ to get the optimal fused prediction $\hat{\boldsymbol{X}}_{fuse}$ with respect to the ground truth $\boldsymbol{X}_{out}$.

## 3. Methodology

This section presents the TIMEFUSE framework for collective time-series forecasting with *sample-level adaptive fusion*. It has two main components: a meta-feature extractor responsible for extracting key statistical and temporal feature descriptors from the input time series, and a learnable model fusor trained to predict the optimal fusion weights based on the meta-features. An overview of the proposed TIMEFUSE framework is shown in Fig. 2.

### 3.1. Time Series Meta-feature Extraction

We start by presenting the foundation of TIMEFUSE: characterizing input time series data by extracting meta-features.

**Why not raw features?** To achieve a dynamic ensemble based on input time-series, one could directly train a fusor using the raw features $\boldsymbol{X}_{in}$. Despite its simplicity, there are several fundamental drawbacks: (i) *Task Dependency*: Forecasting tasks differ greatly in feature dimensions and input lengths, making the fusor task-specific brings increasing training costs while reducing flexibility. (ii) *Risk of Over-*

*fitting*: Raw features, often complex and high-dimensional, can include unnecessary information and noise, raising the likelihood of overfitting, especially with limited meta-training data. (iii) *Lack of Semantic Interpretability*: The complexity of raw features also makes it challenging to intuitively understand and interpret the fusor's behavior.

**Benefits of meta-features.** Drawing on existing research in feature-based time-series analysis (Henderson & Fulcher, 2021; Barandas et al., 2020), we opt to extract key meta-features that simplify the high-dimensional data by retaining only critical statistical and temporal information. This approach brings several benefits: (i) **Versatility**: Meta-features are task-agnostic descriptors that standardize input features across various tasks, enhancing the fusor's adaptability and aiding in generalization through cross-task training. (ii) **Robustness**: Reducing dimensionality helps filter out noise and redundant information, minimizing overfitting risks and improving generalization on new data. We now detail the meta-feature set used. (iii) **Interpretability**: Meta-features provide clear semantics, such as periodicity and spectral density, making it easier to analyze such features and understand how the fusor performs.

**Meta-feature details.** Drawing on prior research in time-series feature extraction, we meticulously crafted a set of meta-features that characterize time-series properties from four distinct perspectives: **(i) Statistical**: Describe the distribution characteristics and basic statistics of the input time series, such as the tendency, dispersion, and symmetry of the data. **(ii) Temporal**: Capture the time dependency and dynamic patterns of the input sequence by depicting how the input sequence evolves, such as long-term trends, recurrent cycles, and rate of change. **(iii) Spectral**: Derived by analyzing the time series in the frequency domain, such as power spectral density and spectral entropy. They highlight the prominence of periodic components and the complexity of the signal. **(iv) Multivariate**: Reflect the relationships among different dimensions in multivariate series, like

*Table 1.* Description of TIMEFUSE meta-features.

| Feature | | Description | Formula |
|---|---|---|---|
| **Statistical** | mean | Average value of the series. | $\frac{1}{T}\sum_{t=1}^{T} x_{in}^{(i)}[t]$ |
| | std | Variability of the series. | $\sqrt{\frac{1}{T}\sum_{t=1}^{T}\left(x_{in}^{(i)}[t]-\text{mean}\right)^2}$ |
| | min | Lowest value in the series. | $\min_t x_{in}^{(i)}[t]$ |
| | max | Highest value in the series. | $\max_t x_{in}^{(i)}[t]$ |
| | skewness | Asymmetry of the series distribution. | $\frac{\frac{1}{T}\sum_{t=1}^{T}(x_{in}^{(i)}[t]-\text{mean})^3}{(\text{std})^3}$ |
| | kurtosis | Sharpness of the series distribution. | $\frac{\frac{1}{T}\sum_{t=1}^{T}(x_{in}^{(i)}[t]-\text{mean})^4}{(\text{std})^4}-3$ |
| **Temporal** | autocorr_mean | Average first-order temporal dependency. | $\text{acf}(x_{in}^{(i)}, \text{lag}=1)$ |
| | stationarity | Ratio of stationary features, determined by | $\text{ADF}(x_{in}^{(i)}) < 0.05$ |
| | roc_mean | Change rate between consecutive steps. | $\frac{1}{T-1}\sum_{t=1}^{T-1}\frac{x_{in}^{(i)}[t+1]-x_{in}^{(i)}[t]}{x_{in}^{(i)}[t]}$ |
| | roc_std | Variability in the change rate. | $\text{std}\left(\frac{x_{in}^{(i)}[t+1]-x_{in}^{(i)}[t]}{x_{in}^{(i)}[t]}\right)$ |
| | autoreg_coef | Mean of the AR(1) coefficients. | $\frac{1}{K}\sum_{i=1}^{K}\phi^{(i)}$ |
| | residual_std | Standard deviations of AR(1) residuals. | $\frac{1}{K}\sum_{i=1}^{K}\text{std}(\epsilon^{(i)})$ |
| **Spectral** | freq_mean | Average energy in the frequency domain. | $\frac{1}{|f|}\sum_f \text{PSD}(x_{in}^{(i)})$ |
| | freq_peak | Dominant periodicity. | $\arg\max_f \text{PSD}(x_{in}^{(i)})$ |
| | spectral_entropy | Complexity in the frequency domain. | $-\sum_f p(f)\log p(f),\ p(f)=\frac{\text{PSD}(f)}{\sum \text{PSD}(f)}$ |
| | spectral_skewness | Asymmetry of the spectral distribution. | $\frac{\sum_f (A(f)-\bar{A})^3}{\left(\sqrt{\sum_f (A(f)-\bar{A})^2}\right)^3}$ |
| | spectral_kurtosis | Sharpness of the spectral distribution. | $\frac{\sum_f (A(f)-\bar{A})^4}{\left(\sum_f (A(f)-\bar{A})^2\right)^2}$ |
| | spectral_variation | Variability of the spectrum over time. | $\frac{1}{T-1}\sum_{t=1}^{T-1}\sqrt{\sum_f (S(f,t+1)-S(f,t))^2}$ |
| **Multivariate** | cov_mean | Average linear relationship strength. | $\text{mean}\left(\text{cov}(x_{in}^{(i)}, x_{in}^{(j)})\right)$ |
| | cov_max | Strongest linear relationship. | $\max \text{cov}(x_{in}^{(i)}, x_{in}^{(j)})$ |
| | cov_min | Weakest linear relationship. | $\min \text{cov}(x_{in}^{(i)}, x_{in}^{(j)})$ |
| | cov_std | Variability in linear relationships. | $\text{std}\left(\text{cov}(x_{in}^{(i)}, x_{in}^{(j)})\right)$ |
| | crosscorr_mean | Average dependency between features. | $\text{mean}\left(\text{corr}(x_{in}^{(i)}, x_{in}^{(j)})\right)$ |
| | crosscorr_std | Variability in dependencies. | $\text{std}\left(\text{corr}(x_{in}^{(i)}, x_{in}^{(j)})\right)$ |

cross-correlation and covariance. These features help describe the dependency structures between variables. We use the average value across all input variables to compute the non-multivariate meta-features. These meta-features are defined and summarized in Table 1. Through ablation studies and comparative analysis, we demonstrated that these 24 features match the performance of a more complex TSFEL (Barandas et al., 2020) feature set containing 165 variables, with significant contributions from each domain to the overall forecasting performance. This indicates that our features offer a comprehensive description of the multifaceted nature of the input time series, effectively supporting the subsequent adaptive model fusion process.

### 3.2. Adaptive Model Fusion with Learnable Fusor

With the meta-characterization of the input time series established, we next explore how to achieve sample-level adaptive model fusion by training a learnable fusor.

**Fusor architecture.** Formally, let $\Psi_\Theta : \mathbb{R}^{d^*} \to \mathbb{R}^k$ be the model fusor parameterized by $\Theta$, and $\Im : \mathbb{R}^{T_{in} \times d} \to \mathbb{R}^{d^*}$ be the meta feature extraction operator where $d^*$ is the number of meta-features. Given a model zoo $\mathcal{F} = \{f_1, f_2, \ldots, f_k\}$ consisting of $k$ models and an input time-series $X_{in} \in \mathbb{R}^{T_{in} \times d}$, our fusor $\Psi_\Theta(\cdot)$ takes the meta-features $x_{in}^* := \Im(X_{in}) \in \mathbb{R}^{d^*}$ as input and outputs a weight vector $w \in \mathbb{R}^k$, where each element $w_i$ represents the contribution of the $i$-th model $f_i$ in the final prediction. We employ a softmax function to normalize $w$ for numerical stability and facilitate training. The final fused forecasting results $\hat{X}_{out}^*$ is then derived by $\hat{X}_{out}^* := \sum_{i=1}^{k} w_i f_i(X_{in})$. Prioritizing interpretability and efficiency, our fusor is a single-layer neural network that learns a linear mapping $\Theta \in \mathbb{R}^{d^* \times k}$ between meta-features and model weights. We note that this simple architecture is sufficient for effectively and accurately capturing the relationship between meta-features and model

capabilities. Using more complex model structures does not enhance performance yet compromises the simplicity that benefits runtime efficiency and interpretability.

**Training objective.** To achieve accurate fused forecasting, we train the fusor to predict the optimal model weight vector that minimizes the forecasting error of the fused output $\hat{X}_{out}^*$ w.r.t the ground truth $X_{out}$. Formally, given the model fusor $\Psi_\Theta(\cdot)$ and meta feature extraction operator $\Im(\cdot)$, the training objective of fusor $\Psi(\cdot; \Theta)$ is:

$$
\arg\min_{\Theta} \mathbb{E}_{(X_{in}, X_{out}) \sim \mathcal{D}_{val}} \mathcal{L}\left(\hat{X}_{out}^*, X_{out}\right)
$$
$$
\text{where } \hat{X}_{out}^* := \sum_{i=1}^{k} \Psi(\Im(X_{in}); \Theta)^{(i)} f_i(X_{in}). \tag{1}
$$

This objective encourages the model fusor to predict optimal weights such that the combined predictions from each model $f_i$ approximate the ground-truth output $X_{out}$. Here the $\mathcal{L}(\cdot, \cdot)$ can be any loss function. In our use case, predictions from different models on the same input can vary significantly due to their distinct architectures, thus we use Huber loss (Huber, 1992) to prevent outlying individual models from affecting the stability of fusor training. The fusor is trained on the held-out validation set $\mathcal{D}_{val}$ (not used during base model training) to optimize fused forecasting performance on unseen data, this also prevents the potential overfitting of the base models from affecting the fusor's generalization capabilities.

**Independent meta-training dataset & pipeline.** As implied by Equation 1, the meta training of TIMEFUSE only depends on the meta-features $\Im(X_{in})$, predictions from base models $f_i(X_{in}), \forall 1 \leq i \leq k$, and the ground truth label $X_{out}$. Consequently, the fusor meta-training is decoupled from the base model training, allowing meta-training dataset to be independently collected and stored. Formally, let $\mathcal{D}^*$ be the meta-training dataset, each sample can be represented by a triplet $(x_{in}^*, \hat{\mathcal{X}}_{out}, X_{out})$, where $x_{in}^* := \Im(X_{in}) \in \mathbb{R}^{d^*}$ is the meta-feature vector of the input time series, $\hat{\mathcal{X}}_{out} := [f_1(X_{in}), f_2(X_{in}), \cdots, f_k(X_{in})] \in \mathbb{R}^{k \times T_{out} \times d}$ is a 3-dimensional tensor storing the base model predictions, and $X_{out}$ is the ground truth. The independence of meta-training also benefits TIMEFUSE 's extensibility: adding new models can be done by simply incorporating their predictions into the tensor $\hat{\mathcal{X}}_{out}$ and training a new fusor on the updated meta-training set. With the above formulations, we can now rewrite the meta-training objective as:

$$
\arg\min_\Theta \mathbb{E}_{(x_{in}^*, \hat{\mathcal{X}}_{out}, X_{out}) \sim \mathcal{D}^*} \mathcal{L}\left(\Psi(x_{in}^*; \Theta)^\top \cdot \hat{\mathcal{X}}_{out}, X_{out}\right). \tag{2}
$$

**Cross-task meta-training with data mixing.** Lastly, benefiting from the task-agnostic meta-features, the fusor can be trained jointly using diverse data from multiple forecasting tasks, thus better generalizing across diverse temporal

*Table 2.* Long-term forecasting results. All results are averaged from 4 prediction lengths {96, 192, 336, 720} with input length 96. A lower MSE or MAE indicates a better prediction, we highlight the **1st** and *2nd* best results. See Table 11 in Appendix for the full results.

| Metric: MSE / MAE | TFuse (Ours) | TXer (2024) | TMixer (2024) | PAttn (2024) | iTF (2024) | TNet (2023) | PTST (2023) | DLin (2023) | FreTS (2023) | FEDF (2022) | NSTF (2022) | LightTS (2022) | InF (2021) | AutoF (2021) |
|---|---|---|---|---|---|---|---|---|---|---|---|---|---|---|
| ETTh1 | **0.430 0.433** | 0.444 0.445 | 0.450 *0.440* | 0.464 0.453 | 0.521 0.489 | 0.460 0.455 | 0.452 0.451 | 0.460 0.457 | 0.481 0.469 | *0.439* 0.458 | 0.647 0.572 | 0.522 0.500 | 1.059 0.808 | 0.545 0.512 |
| ETTh2 | **0.365 0.395** | *0.377 0.401* | 0.393 0.411 | 0.383 0.404 | 0.383 0.407 | 0.409 0.422 | 0.381 0.400 | 0.564 0.519 | 0.529 0.500 | 0.426 0.445 | 0.532 0.492 | 0.654 0.581 | 2.222 1.223 | 0.457 0.465 |
| ETTm1 | **0.369 0.388** | *0.382 0.397* | 0.386 0.401 | 0.397 0.397 | 0.422 0.417 | 0.410 0.418 | 0.388 0.402 | 0.404 0.408 | 0.410 0.418 | 0.616 0.524 | 0.530 0.473 | 0.421 0.421 | 0.843 0.668 | 0.584 0.517 |
| ETTm2 | **0.274 0.319** | *0.274 0.322* | 0.275 0.323 | 0.286 0.331 | 0.292 0.336 | 0.298 0.333 | 0.289 0.334 | 0.355 0.402 | 0.350 0.390 | 0.308 0.351 | 0.518 0.439 | 0.362 0.409 | 1.224 0.849 | 0.337 0.368 |
| Weather | **0.240 0.270** | *0.241 0.271* | 0.244 0.273 | 0.268 0.285 | 0.260 0.281 | 0.259 0.285 | 0.257 0.278 | 0.265 0.317 | 0.258 0.304 | 0.333 0.375 | 0.289 0.309 | 0.269 0.318 | 0.834 0.668 | 0.341 0.382 |
| Electricity | **0.163 0.260** | *0.171* 0.270 | 0.185 0.275 | 0.215 0.294 | 0.180 *0.270* | 0.196 0.297 | 0.204 0.294 | 0.225 0.319 | 0.209 0.296 | 0.222 0.333 | 0.196 0.296 | 0.238 0.339 | 0.358 0.437 | 0.240 0.346 |
| Traffic | **0.419 0.272** | 0.466 0.287 | 0.513 0.306 | 0.555 0.358 | *0.422 0.282* | 0.624 0.331 | 0.482 0.308 | 0.673 0.419 | 0.599 0.376 | 0.711 0.445 | 0.641 0.352 | 0.755 0.472 | 1.315 0.728 | 0.652 0.402 |

patterns and input characteristics. However, implementing cross-task meta-training in practice faces two challenges: (i) The inconsistency in the dimensions of $\hat{\mathcal{X}}_{\text{out}}$ and $\boldsymbol{X}_{\text{out}}$ across tasks prevents the uniform storage and mixed retrieval of data from different tasks. (ii) Imbalanced training sample sizes across tasks can degrade the fusor's performance on tasks with fewer samples. We propose a simple batch-level mixing and balancing strategy to address these issues. Specifically, given $m$ meta-training datasets $\{\mathcal{D}_1^*, \mathcal{D}_2^*, \cdots, \mathcal{D}_m^*\}$ derived from different forecasting tasks, we oversample all datasets to match the size of the largest task ($\max_{0 \leq i \leq m}(|\mathcal{D}_i^*|)$) to balance the data distribution. All oversampled datasets collectively form the joint meta-training dataset $\mathcal{D}_{\text{joint}}^*$. Further, to prevent overfitting issues that might arise from consecutive training on oversampled data from a single task, we alternate training batches from each task within each training step. This dynamic training data mixing promotes the fusor's generalization across various task distributions. Algorithm 1 summarizes the main procedure of TIMEFUSE.

---

**Algorithm 1** TIMEFUSE

**Require:** model zoo $\mathcal{F}$ : $\{f_1, f_2, \ldots, f_k\}$; forecasting dataset $\mathcal{D}$ : $\{(\boldsymbol{X}_{\text{in}}^{(i)}, \boldsymbol{X}_{\text{out}}^{(i)}) \mid 0 \leq i \leq n\}$.
1: **Initialize:** meta-training set $\mathcal{D}^* \leftarrow \emptyset$.
2: **for** $(\boldsymbol{X}_{\text{in}}, \boldsymbol{X}_{\text{out}}) \in \mathcal{D}$ **do**
3:    # *collect meta-training data*
4:    Extract meta-features $\boldsymbol{x}_{\text{in}}^* \leftarrow \Im(\boldsymbol{X}_{\text{in}}) \in \mathbb{R}^{d^*}$;
5:    Prediction tensor $\hat{\mathcal{X}}_{\text{out}} \leftarrow \left[f_i(\boldsymbol{X}_{\text{in}})\right]_{i=1}^{k} \in \mathbb{R}^{k \times T_{\text{in}} \times d}$;
6:    Forecasting ground truth $\boldsymbol{X}_{\text{out}} \in \mathbb{R}^{T_{\text{out}} \times d}$;
7:    Update $\mathcal{D}^* \leftarrow \mathcal{D}^* \cup (\boldsymbol{x}_{\text{in}}^*, \hat{\mathcal{X}}_{\text{out}}, \boldsymbol{X}_{\text{out}})$;
8: **end for**
9: **while** not converged **do**
10:   # *model fusor training*
11:   Update the model fusor $\Psi(\cdot; \Theta)$ with Eq. (2)
   $\arg\min_\Theta \mathbb{E}_{(\boldsymbol{x}_{\text{in}}^*, \hat{\mathcal{X}}_{\text{out}}, \boldsymbol{X}_{\text{out}}) \sim \mathcal{D}^*} \mathcal{L}(\Psi(\boldsymbol{x}_{\text{in}}^*; \Theta)^\top \cdot \hat{\mathcal{X}}_{\text{out}}, \boldsymbol{X}_{\text{out}})$.
12: **end while**
13: **Return:** a TIMEFUSE model fusor $\Psi(\cdot; \Theta)$

---

## 4. Experiments

We conduct extensive experiments to evaluate the effectiveness of TIMEFUSE, covering long-term and short-term fore-

casting, including 16 real-world benchmarks and 13 base forecasting models. The detailed model and experiment configurations are presented in Appendix A.

**Datasets.** For long-term forecasting, we evaluate our method on seven widely-used benchmarks, including the ETT datasets (with 4 subsets: ETTh1, ETTh2, ETTm1, ETTm2), Weather, Electricity, and Traffic, following prior studies (Wang et al., 2024a; Wu et al., 2023; 2021). For short-term forecasting, we use PeMS (Chen et al., 2001), which encompass four public traffic network datasets (PEMS03/04/07/08), along with the EPF (Lago et al., 2021a) datasets for electricity price forecasting on five major power markets (NP, PJM, BE, FR, DE) spanning six years each. A detailed description of these datasets is provided in Table 7.

**Base Models.** To assess TIMEFUSE's ability to unlock superior forecasting performance based on state-of-the-art models, we choose 13 well-known powerful forecasting models as baselines. This includes recently developed advanced forecasting models such as TimeXer (2024c) that exploits exogenous variables, TimeMixer (2024a) with decomposable multiscale mixing, and patching-based models PAttn (2024) and PatchTST (2023). Other strong competitors are also compared against, including iTransformer (2024a), TimesNet (2023), FedFormer (2022), FreTS (2024), DLinear (2023a), Non-stationary Transformer (2022), LightTS (2022), InFormer (2021), and AutoFormer (2021).

**Setup.** To ensure a fair comparison, we employed the `TSLib` (Wang et al., 2024b) toolkit to implement all base models, using the optimal hyperparameters provided in the official training configurations. We trained all base models using L2 loss on the training sets of each task, and collected meta-training data on the validation sets. For each prediction length, we jointly trained a fusor using data from different tasks and reported the performance of fused predictions on the test set. More reproducibility details are in Appendix A.

### 4.1. Main Results

**Long-term forecasting.** As shown in Table 2, by dynamically leveraging and combining the strengths of different base models, TIMEFUSE consistently outperforms the state-of-the-art individual models across all tasks, cov-

*Table 3.* Short-term forecasting results on PEMS datasets, averaged from 3 prediction lengths {6, 12, 24} with input length 96. Lower MAE/RMSE/MAPE indicates better prediction, we highlight the **1st** and *2nd* best results. See Table 12 in Appendix for the full results.

| Datasets | Metric | TFuse (Ours) | TXer (2024) | TMixer (2024) | PAttn (2024) | iTF (2024) | TNet (2023) | PTST (2023) | DLin (2023) | FreTS (2023) | FEDF (2022) | NSTF (2022) | LightTS (2022) | InF (2021) | AutoF (2021) |
|---|---|---|---|---|---|---|---|---|---|---|---|---|---|---|---|
| **PEMS03** | MAE | **16.005** | 17.900 | 18.283 | 19.222 | *17.420* | 19.349 | 19.499 | 22.028 | 18.634 | 31.446 | 20.337 | 18.140 | 20.842 | 35.842 |
| | RMSE | **24.971** | *27.206* | 28.435 | 30.442 | 27.385 | 30.715 | 30.722 | 35.324 | 29.437 | 44.941 | 31.677 | 27.624 | 32.398 | 52.636 |
| | MAPE | **0.135** | 0.170 | 0.159 | 0.157 | *0.146* | 0.168 | 0.164 | 0.196 | 0.153 | 0.297 | 0.179 | 0.168 | 0.179 | 0.341 |
| **PEMS04** | MAE | **20.268** | 22.513 | 23.631 | 25.925 | 23.245 | 23.028 | 26.591 | 27.743 | 24.817 | 37.439 | 22.855 | *22.237* | 23.155 | 42.046 |
| | RMSE | **32.207** | *34.199* | 36.922 | 40.859 | 36.913 | 36.076 | 40.871 | 42.768 | 38.646 | 53.473 | 35.973 | 34.702 | 37.252 | 59.421 |
| | MAPE | **0.167** | 0.210 | 0.201 | 0.205 | *0.190* | 0.197 | 0.224 | 0.253 | 0.204 | 0.316 | 0.195 | 0.194 | 0.195 | 0.351 |
| **PEMS07** | MAE | **22.988** | 26.695 | 26.489 | 28.946 | *25.934* | 27.977 | 30.641 | 33.365 | 28.304 | 53.631 | 28.978 | 26.005 | 30.211 | 62.726 |
| | RMSE | **36.241** | *39.512* | 41.222 | 44.996 | 40.938 | 44.423 | 45.864 | 50.174 | 43.382 | 73.015 | 45.328 | 39.791 | 48.006 | 84.144 |
| | MAPE | **0.110** | 0.142 | 0.129 | 0.138 | *0.126* | 0.138 | 0.157 | 0.176 | 0.139 | 0.278 | 0.145 | 0.131 | 0.148 | 0.348 |
| **PEMS08** | MAE | **16.468** | 18.613 | 19.114 | 19.920 | *17.978* | 20.367 | 20.623 | 22.513 | 19.172 | 35.569 | 19.712 | 18.103 | 22.870 | 36.078 |
| | RMSE | **25.882** | 27.960 | 29.601 | 31.494 | 28.661 | 31.869 | 31.946 | 35.039 | 30.249 | 50.437 | 30.643 | *27.875* | 35.085 | 50.548 |
| | MAPE | **0.103** | 0.126 | 0.125 | 0.121 | *0.111* | 0.131 | 0.132 | 0.143 | 0.118 | 0.223 | 0.128 | 0.117 | 0.150 | 0.246 |

*Table 4.* Short-term forecasting results on the EPF datasets with predict length 24 and input length 168 following Wang et al. (2024c).

| Metric: MSE / MAE | TFuse (Ours) | | TXer (2024) | | TMixer (2024) | | PAttn (2024) | | iTF (2024) | | TNet (2023) | | PTST (2023) | | DLin (2023) | | FreTS (2023) | | FEDF (2022) | | NSTF (2022) | | LightTS (2022) | | InF (2021) | | AutoF (2021) | |
|---|---|---|---|---|---|---|---|---|---|---|---|---|---|---|---|---|---|---|---|---|---|---|---|---|---|---|---|---|---|
| NP | **0.229** | **0.260** | *0.243* | *0.271* | 0.264 | 0.290 | 0.275 | 0.299 | 0.260 | 0.287 | 0.244 | 0.281 | 0.276 | 0.294 | 0.297 | 0.311 | 0.311 | 0.321 | 0.331 | 0.365 | 0.276 | 0.291 | 0.299 | 0.318 | 0.287 | 0.326 | 0.417 | 0.401 |
| PJM | **0.085** | **0.185** | 0.097 | *0.194* | 0.118 | 0.229 | 0.110 | 0.215 | *0.097* | 0.197 | 0.114 | 0.216 | 0.105 | 0.209 | 0.104 | 0.211 | 0.122 | 0.231 | 0.133 | 0.239 | 0.132 | 0.235 | 0.109 | 0.212 | 0.132 | 0.216 | 0.138 | 0.251 |
| BE | **0.378** | **0.241** | *0.379* | *0.242* | 0.415 | 0.276 | 0.396 | 0.264 | 0.404 | 0.277 | 0.401 | 0.274 | 0.404 | 0.261 | 0.459 | 0.315 | 0.437 | 0.311 | 0.474 | 0.332 | 0.396 | 0.253 | 0.444 | 0.297 | 0.543 | 0.369 | 0.481 | 0.311 |
| FR | **0.386** | **0.198** | *0.396* | *0.212* | 0.444 | 0.245 | 0.448 | 0.255 | 0.420 | 0.228 | 0.427 | 0.230 | 0.409 | 0.219 | 0.420 | 0.254 | 0.403 | 0.236 | 0.456 | 0.273 | 0.465 | 0.240 | 0.473 | 0.258 | 0.479 | 0.264 | 0.559 | 0.292 |
| DE | **0.429** | **0.407** | *0.458* | *0.422* | 0.469 | 0.437 | 0.472 | 0.434 | 0.466 | 0.435 | 0.472 | 0.433 | 0.527 | 0.457 | 0.508 | 0.457 | 0.511 | 0.452 | 0.626 | 0.529 | 0.506 | 0.455 | 0.489 | 0.442 | 0.593 | 0.494 | 0.629 | 0.517 |

ering a large variety of time series with varying frequencies, numbers of variables, and real-world application domains. We also note that the optimal individual model varies across different tasks and metrics: while TimeXer (Wang et al., 2024a) excels in most long-term forecasting tasks by modeling exogenous variables, other models like TimeMixer (Wang et al., 2024a), iTransformer (Liu et al., 2024a), and FedFormer (Zhou et al., 2022) still emerge as top performers in certain tasks. In contrast, the proposed TIMEFUSE consistently achieves superior results across different tasks by integrating their advantages, surpassing the *task-specific best models*. Compared with these SOTA models, TIMEFUSE achieved an average MSE reduction of 3.61%/6.88%/11.77%/8.39% across seven datasets w.r.t TXer/Tmixer/PAttn/iTF, respectively, demonstrating the efficacy and versatility of the proposed TIMEFUSE framework.

**Short-term forecasting.** TIMEFUSE also demonstrates superb performance in short-term forecasting tasks. As shown in Table 3, on the PEMS datasets, while iTransformer achieves the overall best performance for this task, models like TimeXer and the MLP-based LightTS also emerge as top performers in many cases. By dynamically leveraging their respective strengths, TIMEFUSE achieves universally enhanced performance, consistently delivering optimal forecasting results across all scenarios. Compared to the top three individual models: TimeXer, iTransformer, and LightTS, TIMEFUSE achieves an average MAPE reduction of 20.46%, 9.89%, and 15.37%, respectively. On the EPF datasets reported in Table 4, while TimeXer exhibits significant advantages at the dataset level, TIMEFUSE still consistently achieves more accurate forecasting across all five tasks by integrating the capabilities of different models

via sample-level adaptive fusion. This further underscores the unique strengths of different models and the importance of combining them with adaptive model fusion.

**Comparison with ensemble methods.** We further compared the performance of TIMEFUSE with commonly used mean/median ensemble for heterogeneous model fusion. To fully unlock the potential of static ensembling, we further adopt a top-$k$ model selection strategy based on the validation performance, and test the mean/median ensemble of the top-$k$ models. Figure 3 shows the results on long-term forecasting datasets. We observed that: (1) the adaptive fusion strategy of TimeFuse consistently outperformed static ensemble strategies across all tasks, even though the static ensembles also equipped a model filtering strategy based on the validation performance. (2) The optimal model set for static ensemble differs significantly across different datasets, e.g., mean ensemble achieves the best result with 10 models included on the ETTh1 dataset, while only the top 2 models are needed on the traffic dataset. In practice, finding the optimal model set for each dataset is cumbersome, highlighting the limitations of static ensemble strategies. In contrast, TIMEFUSE dynamically learns the optimal ensemble strategy for each sample (and thus dataset), which we will discuss further in the following section.

### 4.2. Further Analysis

**TIMEFUSE learns adaptive fusion strategies.** To intuitively grasp how TIMEFUSE learns different fusion strategies across various datasets and samples, Figure 4 shows the average model fusion weights on different datasets given by TIMEFUSE, with variation bars indicating standard de-

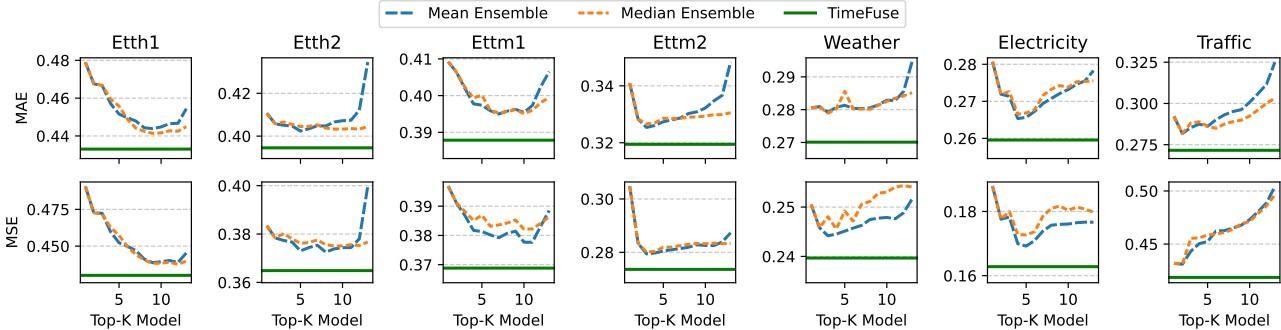

*Figure 3.* Comparison between TIMEFUSE and static ensemble methods with validation top-k model filtering.

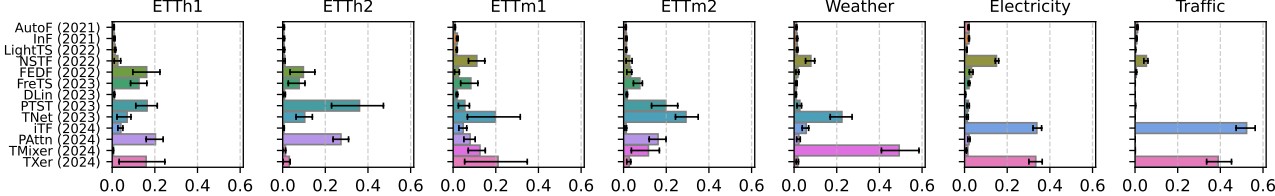

*Figure 4.* Visualization of the average model fusion weights by TIMEFUSE across datasets, with variation bars showing standard deviations between samples within the dataset. TIMEFUSE adaptively produces diverse ensemble strategies, effectively supporting tasks that benefit from either a broad model ensemble (e.g., ETTh1/m1) or a more selective integration of specific models (e.g., Electricity/Traffic).

viations among samples within each dataset. It can be observed that TIMEFUSE adaptively generates diverse ensemble strategies. Notably, on some datasets (e.g., ETTh1/m1) TIMEFUSE opts for a dynamic ensemble of a variety of models, while on others (e.g., Electricity/Traffic), it selectively ensembles only a few specific models. This echoes the observations in Figure 3, where some datasets benefit from a broad model ensemble and others from a more selective integration, further demonstrating TIMEFUSE's ability to learn effective and adaptive ensemble strategies.

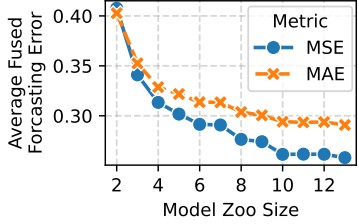

*Figure 5.* TIMEFUSE achieves increasingly better forecasting performance as more base models are included in the model zoo.

**TIMEFUSE improves with a more diverse model zoo.** To assess whether TIMEFUSE can leverage emerging models to further enhance its capabilities, we check how the fused forecasting performance changes as new models are introduced into the model zoo. Specifically, we test TIMEFUSE on long-term forecasting datasets with a prediction length of 96, starting with a model zoo that contains only Informer and Autoformer, with newer models added sequentially as per the order in Table 2 (from right to left). Figure 5 shows the average MAE and MSE on 7 datasets. We can observe that TimeFuse achieves increasingly better forecasting performance as more diverse base models are integrated. This indicates that TIMEFUSE benefits from the capabilities of

new models and the added diversity they bring to the model zoo, allowing for continuous performance improvement as more advanced forecasting models emerge in the future.

**Zero-shot generalization to unseen datasets.** As mentioned before, TIMEFUSE operates on top of task-agnostic meta-features. This allows it to be jointly trained on various datasets to enhance its generalizability, and further, to perform inference on any datasets, even unseen during meta-training (i.e., zero-shot generalization). To verify this, we evaluate zero-shot TIMEFUSE on both long- (predict length 96) and short-term (PEMS with predict length 6) forecasting tasks. For each target dataset, we trained a fusor with all other datasets, and then tested it on the target dataset. As shown in Table 5, zero-shot TIMEFUSE still outperforms the best individual model in most cases, demonstrating robust zero-shot generalization performance.

*Table 5.* Zero-shot performance of TIMEFUSE.

| Dataset | Normal TIMEFUSE | | Zero-shot TIMEFUSE | | Best Individual Model | |
|---|---|---|---|---|---|---|
| | MSE | MAE | MSE | MAE | MSE | MAE |
| **ETTh1** | **0.3667** | **0.3911** | *0.3707* | *0.3961* | 0.3770 | 0.3981 |
| **ETTh2** | **0.2803** | **0.3338** | *0.2817* | *0.3348* | 0.2854 | 0.3376 |
| **ETTm1** | **0.3060** | **0.3505** | *0.3078* | *0.3505* | 0.3178 | 0.3563 |
| **ETTm2** | **0.1701** | **0.2542** | 0.1721 | 0.2568 | *0.1712* | *0.2560* |
| **Weather** | **0.1546** | **0.2046** | *0.1552* | 0.2048 | 0.1574 | *0.2047* |
| **Electricity** | **0.1334** | **0.2312** | *0.1355* | *0.2336* | 0.1405 | 0.2406 |
| **Traffic** | **0.3881** | **0.2552** | *0.3907* | *0.2602* | 0.3936 | 0.2686 |

| Datasets | Normal TIMEFUSE | | | Zero-shot TIMEFUSE | | | Best Individual Model | | |
|---|---|---|---|---|---|---|---|---|---|
| | MAE | RMSE | MAPE | MAE | RMSE | MAPE | MAE | RMSE | MAPE |
| **PEMS03** | **14.266** | **22.235** | **0.118** | *14.273* | *22.257* | *0.118* | 14.941 | 23.314 | 0.122 |
| **PEMS04** | **18.936** | **30.338** | **0.154** | *19.023* | *30.467* | *0.154* | 20.073 | 31.934 | 0.168 |
| **PEMS07** | **21.205** | **33.432** | **0.099** | *21.297* | *33.523* | *0.100* | 22.243 | 34.950 | 0.109 |
| **PEMS08** | **14.930** | **23.570** | **0.090** | *14.933* | *23.571* | *0.090* | 15.661 | 24.577 | 0.095 |

*Table 6.* Ablation and comparative study of meta-features.

| Datasets | | Ours | TSFEL | Ablation Feature Set | | | |
|---|---|---|---|---|---|---|---|
| | | | | Statistical | Temporal | Spectral | Multivariate |
| Long-term | ETTh1 | 0.3911 | 0.3938 | 0.3978 | 0.4007 | 0.4010 | 0.3986 |
| | ETTh2 | 0.3338 | 0.3343 | 0.3406 | 0.3418 | 0.3393 | 0.3409 |
| | ETTm1 | 0.3505 | 0.3516 | 0.3552 | 0.3578 | 0.3558 | 0.3575 |
| | ETTm2 | 0.2542 | 0.2550 | 0.2578 | 0.2579 | 0.2566 | 0.2553 |
| | Weather | 0.2046 | 0.2042 | 0.2074 | 0.2088 | 0.2096 | 0.2081 |
| | Electricity | 0.2312 | 0.2327 | 0.2363 | 0.2367 | 0.2364 | 0.2361 |
| | Traffic | 0.2552 | 0.2572 | 0.2590 | 0.2626 | 0.2661 | 0.2607 |
| | Avg. | **0.2887** | *0.2898* | 0.2934 | 0.2952 | 0.2950 | 0.2939 |
| Short-term | PEMS03 | 14.266 | 14.270 | 14.521 | 14.584 | 14.593 | 14.547 |
| | PEMS04 | 18.936 | 18.939 | 19.235 | 19.485 | 19.410 | 19.346 |
| | PEMS07 | 21.205 | 21.308 | 21.696 | 21.902 | 21.894 | 21.905 |
| | PEMS08 | 14.930 | 14.950 | 15.189 | 15.371 | 15.318 | 15.264 |
| | Avg. | **17.334** | *17.367* | 17.660 | 17.836 | 17.804 | 17.765 |

**Ablation and comparative study of meta-features.** We also conducted experiments to validate the effectiveness of the meta-features used. We compare our 24-variable meta-feature set with a more complex TSFEL (Barandas et al., 2020) feature set comprising 165 variables, and perform an ablation study by removing features of each domain from the meta-feature set. The MAE of long/short-term forecasting (with predict length 96/6) is shown in Table 6. It can be observed that our meta-feature set offers a comprehensive description of the multifaceted nature of the input time series, matching the performance of the complex TSFEL set, with significant contributions from each domain.

**Visualization of the learned fusor weights.** Finally, we note that the fusor essentially learns a mapping of how specific input temporal properties (e.g., stationarity, spectral complexity) correspond to the strengths of different models. We visualized the learned fusor weights on long-term forecasting datasets in Figure 6 (showing a subset of meta-features for clarity). Take the meta-feature "stationarity" as an example, it corresponds to large negative weights for the non-stationary transformer, suggesting that *this model is weighted more heavily when inputs show lower stationarity*, aligning with its ability to handle non-stationary dynamics. Similarly, TimeMixer's advantage in modeling complex frequency patterns through multi-scale mixing is also reflected in the learned fusor weights. We note that these advantages are relative: they indicate a model's relative strength in handling samples with specific properties compared to others in the model zoo. Nonetheless, such understanding can help users grasp the relative strengths and weaknesses of different models and provide insights for further improvement.

**Comparison with AutoML baselines and more analysis.** We include additional experiments in Appendix B to further validate the robustness and practicality of TIMEFUSE. These include comparisons with advanced ensemble strategies such as portfolio-based, zero-shot, and AutoML-driven methods, as well as pretrained foundation models (Sec-

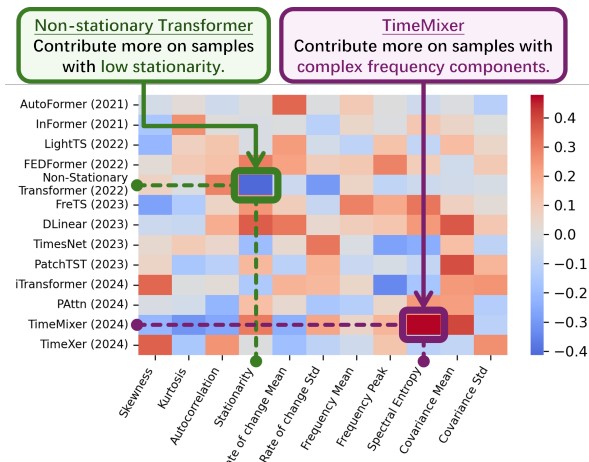

*Figure 6.* Visualization of the learned fusor weights.

tion B.3). We also report the inference efficiency of the fusor relative to base models (Section B.1), and evaluate performance under up to 512 long input horizons (Section B.2). The results demonstrate that TIMEFUSE remains effective, efficient, and adaptable across diverse forecasting scenarios.

## 5. Related Works

**Time-series Forecasting.** Time-series forecasting is a pivotal research area with rich real-world applications (Lim & Zohren, 2021). Numerous time-series modeling techniques have been proposed, each with its unique focus. Traditional statistical methods (Anderson, 1976) can handle periodic trends but struggle with complex nonlinear dynamics. Later works based on recurrent (Lai et al., 2018) and convolutional (Franceschi et al., 2019) neural networks can model more complex temporal patterns but still have difficulty with long-range dependencies due to the Markovian assumption or limited receptive field. TimesNet (Wu et al., 2023) addresses this by transforming 1D series into 2D formats, enhancing pattern recognition over distances. Meanwhile, Transformer-based models like PatchTST (Nie et al., 2023) and iTransformer (Liu et al., 2024a) leverage self-attention to model long-range dependencies. More recent studies further suggest improving forecasts by multiscale mixing (Wang et al., 2024a) or integrating exogenous variables (Wang et al., 2024c) or multimodal knowledge (Li et al., 2025). Despite these advancements, our sample-level inspection shows that no single model excels universally, prompting research into leveraging the diverse strengths of various models for enhanced joint forecasting.

**Ensemble Learning.** Ensemble learning is a generic strategy to get robust predictions by aggregating outputs from multiple models (Mienye & Sun, 2022). Typical ensemble methods often employ quickly trainable weak base learners like decision trees (Sagi & Rokach, 2018). Research on time-series forecasting ensembles has been limited due to the complexity of time-series data modeling. To name a

few, Kourentzes et al. (2014) and Oliveira & Torgo (2015) explore the potential of mean/median and bagging ensembles in time series forecasting tasks. Yu et al. (2017) implement additive forecasting using multiple MLPs across varied feature sets, a method similar to random subspace ensembles (Ho, 1998), and Choi & Lee (2018) further integrates multiple LSTMs. However, these studies are confined to a single-model architecture, focusing on training multiple same-type models based on different dataset views for a static homogeneous ensemble. Our work, in contrast, enables dynamic heterogeneous ensemble of models with various architectures. By employing sample-level adaptive model fusion, we dynamically integrate the strengths of different models to achieve superior joint forecasting.

## 6. Conclusion

We present TIMEFUSE, a versatile framework for ensemble time-series forecasting with sample-level adaptive model fusion. It learns and selects the optimal model combination based on the characteristics of the input time series in an interpretable and adaptive manner, thereby unlocking more accurate predictions. In all our experiments, TIMEFUSE consistently achieved new state-of-the-art performance by dynamically leveraging different models' strengths at test time, highlighting its strong capability as a versatile tool for tackling complex time series forecasting challenges.

## Impact Statement

This paper presents a novel learning-based ensemble time-series forecasting framework, whose goal is to achieve more accurate forecasting by adaptively fusing different models for each input time series at test time. Like other research focused on time series forecasting, our work has potential social impacts, but none of which we feel must be highlighted here.

## Acknowledgements

This work is supported by NSF (2324770, 2117902), MIT-IBM Watson AI Lab, and IBM-Illinois Discovery Accelerator Institute. The content of the information in this document does not necessarily reflect the position or the policy of the Government, and no official endorsement should be inferred. The U.S. Government is authorized to reproduce and distribute reprints for Government purposes notwithstanding any copyright notation here on.

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

# Appendix

# A. Reproducibility Details

## A.1. Dataset Descriptions

**Long-term forecasting datasets.** We conduct long-term forecasting experiments on 7 well-established real-world datasets, including: (1) **ETT** (Zhou et al., 2021) contains 2-year electricity transformer temperature datasets collected from two separate counties in China. It contains four subsets where ETTh1 and ETTh2 are hourly recorded, and ETTm1 and ETTm2 are recorded every 15 minutes. Each data point records the oil temperature and 6 power load features. The train/val/test is 12/4/4 months. (2) **Weather** (Zhou et al., 2021) records 21 meteorological factors collected every 10 minutes from the Weather Station of the Max Planck

Biogeochemistry Institute in 2020. (3) **Electricity** (ecl) includes hourly electricity consumption data for 2 years from 321 clients. The train/val/test is 15/3/4 months. (4) **Traffic** (Wu et al., 2023) records hourly road occupancy rates measured by 862 sensors of San Francisco Bay area freeways.

**Short-term forecasting datasets.** We test the short-term forecasting performance on the PEMS (tra) dataset for traffic flow forecasting following TimeMixer (Wang et al., 2024a), and EPF (Lago et al., 2021b) datasets for electricity price prediction following TimeXer (Wang et al., 2024c). The PEMS dataset contains four public traffic network datasets (PEMS03, PEMS04, PEMS07, PEMS08) that record the traffic flow data collected by sensors spanning the freeway system in the State of California. The EPF dataset contains five datasets representing five different day-ahead electricity markets (NP: Nord Pool; PJM: Pennsylvania-New Jersey-Maryland; BE: Belgium; FR: France; DE: German) spanning six years each.

The detailed statistics of each dataset are given in Table 7.

## A.2. Evaluation Metric Details

Regarding metrics, we utilize the mean square error (MSE) and mean absolute error (MAE) for all long-term forecasting tasks and EPF datasets for short-term forecasting following (Wang et al., 2024c). For the PEMS dataset, we follow the metrics of TimeMixer (Wang et al., 2024a) to report the mean absolute error (MAE), root mean squared error (RMSE), and mean absolute percentage error (MAPE) on the unnormalized test set. Let $\boldsymbol{X}_{\text{out}}^{(i)} \in \mathbb{R}^{T_{\text{out}} \times d}$ and $\hat{\boldsymbol{X}}_{\text{out}}^{(i)} \in \mathbb{R}^{T_{\text{out}} \times d}$ be the ground truth and model prediction results of the $i$-th sample, these metrics are computed as follows:

$$\text{MSE} = \sum_{i=1}^{F} (\boldsymbol{X}_{\text{out}}^{(i)} - \hat{\boldsymbol{X}}_{\text{out}}^{(i)})^2,$$

$$\text{MAE} = \sum_{i=1}^{F} |\boldsymbol{X}_{\text{out}}^{(i)} - \hat{\boldsymbol{X}}_{\text{out}}^{(i)}|,$$

$$\text{RMSE} = (\text{MSE})^{\frac{1}{2}},$$

$$\text{MAPE} = \frac{100}{F} \sum_{i=1}^{F} \frac{|\boldsymbol{X}_{\text{out}}^{(i)} - \hat{\boldsymbol{X}}_{\text{out}}^{(i)}|}{|\boldsymbol{X}_{\text{out}}^{(i)}|}.$$

## A.3. Implementation Details

**Base Forecasting Models.** To assess TIMEFUSE's ability to unlock superior forecasting performance based on the state-of-the-art time-series models, we chose 13 well-known powerful forecasting models as our baselines. They include recently developed advanced forecasting models such as TimeXer (Wang et al., 2024c) that exploits exoge-

*Table 7.* Dataset detailed descriptions. The dataset size is organized in (Train, Validation, Test).

| Tasks | Dataset | #Variates | Input Length | Predict Length | Dataset Size | Frequency | Description |
|-------|---------|-----------|--------------|----------------|--------------|-----------|-------------|
| Long-term Forecasting | ETTm1 | 7 | 96 | {96, 192, 336, 720} | (34465, 11521, 11521) | 15 min | Electricity Transformer Temperature |
| | ETTm2 | 7 | 96 | {96, 192, 336, 720} | (34465, 11521, 11521) | 15 min | Electricity Transformer Temperature |
| | ETTh1 | 7 | 96 | {96, 192, 336, 720} | (8545, 2881, 2881) | 1 hour | Electricity Transformer Temperature |
| | ETTh2 | 7 | 96 | {96, 192, 336, 720} | (8545, 2881, 2881) | 1 hour | Electricity Transformer Temperature |
| | Electricity | 321 | 96 | {96, 192, 336, 720} | (18317, 2633, 5261) | 1 hour | Electricity Consumption |
| | Weather | 21 | 96 | {96, 192, 336, 720} | (36792, 5271, 10540) | 10 min | Climate Feature |
| | Traffic | 862 | 96 | {96, 192, 336, 720} | (12185, 1757, 3509) | 1 hour | Road Occupancy Rates |
| Short-term Forecasting | PEMS03 | 358 | 96 | {6, 12, 24} | (15617, 5135, 5135) | 5min | Traffic Flow |
| | PEMS04 | 307 | 96 | {6, 12, 24} | (10172, 3375, 3375) | 5min | Traffic Flow |
| | PEMS07 | 883 | 96 | {6, 12, 24} | (16911, 5622, 5622) | 5min | Traffic Flow |
| | PEMS08 | 170 | 96 | {6, 12, 24} | (10690, 3548, 265) | 5min | Traffic Flow |
| | EPF-NP | 1 | 168 | 24 | (36500, 5219, 10460) | 1 hour | Electricity Price |
| | EPF-PJM | 1 | 168 | 24 | (36500, 5219, 10460) | 1 hour | Electricity Price |
| | EPF-BE | 1 | 168 | 24 | (36500, 5219, 10460) | 1 hour | Electricity Price |
| | EPF-FR | 1 | 168 | 24 | (36500, 5219, 10460) | 1 hour | Electricity Price |
| | EPF-DE | 1 | 168 | 24 | (36500, 5219, 10460) | 1 hour | Electricity Price |

nous variables to enhance forecasting, TimeMixer (Wang et al., 2024a) with decomposable multiscale mixing for modeling distinct patterns in different sampling scales, patching-based transformer models like PAttn (Tan et al., 2024) and PatchTST (Nie et al., 2023) that model the global dependencies over temporal tokens of time series, and iTransformer (Liu et al., 2024a) that applies the attention on the inverted dimensions to capture multivariate correlations between variate tokens. Other strong competitive models including TimesNet (Wu et al., 2023), FedFormer (Zhou et al., 2022), FreTS (Yi et al., 2024), DLinear (Zeng et al., 2023a), Non-stationary Transformer (Liu et al., 2022), LightTS (Zhang et al., 2022), InFormer (Zhou et al., 2021), and AutoFormer (Wu et al., 2021). To ensure a fair comparison, we employed the `TSLib` (Wang et al., 2024b) toolkit[1] to implement all base models, the optimal hyperparameters for each model-dataset pair are used if provided in the official training configurations. Unless otherwise specified in the training configuration, all models are trained for 10 epochs using an ADAM optimizer (Kingma, 2014) with L2 loss, we also perform early stopping with a patience of 3 based on validation set loss to prevent overfitting. All experiments are conducted on a single NVIDIA A100 80GB GPU.

**TIMEFUSE details.** We use Pytorch (Paszke et al., 2019) to implement the fusor, which is a single-layer neural network that learns a linear mapping $\Theta \in \mathbb{R}^{d^* \times k}$ from meta-features to model weights. For all long-term forecasting tasks (i.e., ETTh1/h2/m1/m2, Weather, Electricity, Traffic), we collect meta-training data from their validation sets as

[1] https://github.com/thuml/Time-Series-Library

described in Section 3.2, then jointly train a single fusor and test its dynamic ensemble prediction performance on each task's test set. Similarly, we conduct joint fusor training and testing on the PEMS and EPF datasets for short-term forecasting. The fusor is optimized using the ADAM (Kingma, 2014) optimizer and Huber loss, with a batch size of 32 and a learning rate of 1e-3.

## B. Additional Experiments and Analysis

### B.1. Inference Efficiency

We benchmark the runtime efficiency of the TIMEFUSE fusor relative to its base models. Table 8 reports batch inference time (batch size 32, prediction length 96) across long-term forecasting datasets. It can be observed that thanks to its simple architecture (a single-layer neural network), the fusor introduces negligible overhead. It is significantly faster than transformer-based models and is on par with lightweight models like `DLinear`. This makes TIMEFUSE practical for latency-sensitive applications and large-scale deployments.

### B.2. Forecasting with Extended Input Horizons

We assess how TIMEFUSE performs under longer input lookback lengths ($L = 336$ and $L = 512$). Table 9 shows the forecasting results across seven long-term datasets. We observe that while models such as `PatchTST` and `PAttn` benefit from longer input sequences, others like `TimeXer`, `iTransformer`, or `TimesNet` show mixed or degraded performance. In contrast, TIMEFUSE consistently improve or maintain strong performance across all input lengths by dynamically leveraging each model's strengths. This

*Table 8.* Batch inference time (in milliseconds) of the TimeFuse fusor and each base model on long-term forecasting datasets (predict length 96, batch size 32). All runtime results are collected from a Linux server with NVIDIA V100-32GB GPU.

| Datasets | #Variates | TimeFuse (Ours) | TXer (2024) | TMixer (2024) | PAttn (2024) | iTF (2024) | TNet (2023) | PTST (2023) | DLin (2023) | FreTS (2023) | FEDF (2022) | NSTF (2022) | LightTS (2022) | InF (2021) | AutoF (2021) |
|---|---|---|---|---|---|---|---|---|---|---|---|---|---|---|---|
| ETTh1 | 7 | | 2.37 | 7.77 | 1.72 | 2.25 | 17.30 | 1.96 | 0.80 | 1.41 | 52.52 | 7.05 | 1.29 | 13.04 | 35.07 |
| ETTh2 | 7 | | 2.41 | 7.18 | 1.58 | 2.42 | 23.11 | 3.13 | 0.53 | 1.11 | 50.86 | 12.68 | 1.20 | 6.83 | 35.41 |
| ETTm1 | 7 | | 2.42 | 6.59 | 1.67 | 2.32 | 26.90 | 1.61 | 0.90 | 1.14 | 51.48 | 13.58 | 1.27 | 6.86 | 37.59 |
| ETTm2 | 7 | 0.145 | 2.37 | 8.71 | 1.63 | 2.16 | 20.75 | 3.24 | 0.64 | 1.20 | 51.38 | 12.93 | 1.39 | 7.04 | 28.35 |
| electricity | 321 | | 6.83 | 10.22 | 2.22 | 3.61 | 1237.25 | 3.16 | 0.78 | 1.87 | 51.47 | 65.95 | 1.57 | 13.23 | 39.84 |
| weather | 21 | | 2.32 | 9.41 | 2.10 | 3.55 | 22.02 | 2.96 | 0.78 | 1.56 | 52.93 | 13.01 | 1.56 | 8.38 | 40.43 |
| traffic | 862 | | 5.99 | 9.39 | 2.57 | 4.61 | 1034.37 | 3.03 | 0.79 | 1.76 | 53.25 | 14.46 | 1.65 | 8.85 | 41.15 |

demonstrates its robustness to changes in temporal context and model behaviors.

### B.3. Comparison with AutoML and Advanced Ensemble Baselines

To further validate the robustness of TIMEFUSE, we compare its performance with several strong baselines:

**Advanced ensemble strategies:** We include three representative methods: (1) *Forward selection*, a greedy Caruana-style ensemble (Caruana et al., 2004) that sequentially adds models to minimize validation loss; (2) *Portfolio-based ensemble*, which selects a subset of models based on overall validation performance and averages their predictions (Feurer et al., 2022); (3) *Zero-shot ensemble*, which computes similarity between input meta-features and training tasks to weight base models without retraining (Feurer et al., 2022).

**AutoML ensemble (AutoGluon-TimeSeries):** A fully automated ensemble system that combines 24 diverse base models using multi-layer stacking and weighted averaging, optimized through validation scores (Shchur et al., 2023). It supports probabilistic forecasting but is limited to univariate targets.

**Foundation model (Chronos-Bolt-Base):** A large pretrained transformer model for univariate time-series forecasting (Ansari et al., 2024). We evaluate both its zero-shot performance and a fine-tuned version using the AutoGluon API. Note that Chronos is trained with short prediction horizons and does not natively support multivariate or long-range forecasting tasks.

All methods are evaluated across 16 datasets from three task types: long-term multivariate, short-term multivariate, and short-term univariate forecasting. As summarized in Table 10, TIMEFUSE consistently achieves the best average performance across all three categories.

**Static nature of traditional ensembles.** Traditional ensemble strategies such as forward selection and portfolio-based ensembling determine static model weights based on aggregate validation performance across the training dataset. While effective in capturing coarse-level model strengths, they fail to account for input-dependent variability at inference time. Consequently, their fusion strategies cannot adapt to diverse temporal patterns or regime shifts in test data. In contrast, TIMEFUSE dynamically adjusts fusion weights for each input instance using meta-features, enabling finer-grained modeling and improved generalization.

**Limitations of zero-shot ensembling.** Zero-shot ensemble methods attempt to address adaptivity by computing similarity between new inputs and previously seen datasets using meta-features. However, their conditioning granularity is limited to dataset-level similarity, rather than per-sample adaptivity, and their effectiveness depends heavily on the diversity and coverage of training tasks.

**Chronos-Bolt and the challenge of generalization.** Chronos-Bolt, a pretrained foundation model, is optimized for short-horizon univariate tasks with a maximum prediction length of 64. While fine-tuning improves performance on some benchmarks, its architectural constraints and training objectives make it ill-suited for long-term or multivariate forecasting. In our evaluations, Chronos yields unstable or degenerate predictions (e.g., extremely high MSE on ETTm2 and Traffic) under such settings.

**AutoGluon and inference inefficiency.** AutoGluon-TimeSeries, while offering flexible ensembling over a wide model zoo, is currently restricted to univariate forecasting. Its ensemble predictions are computed independently for each variate, which leads to prohibitive computational costs in high-dimensional settings. Additionally, many of its constituent models (e.g., Prophet, ARIMA) do not support GPU acceleration, resulting in hours-long inference times for large datasets such as Electricity (321 variates) and Traffic (862 variates). These limitations hinder its applicability in real-time or resource-constrained scenarios.

Taken together, these results highlight the advantage of TIMEFUSE as a lightweight, flexible, and input-aware ensemble framework that generalizes well across forecasting settings with varying dimensionality, horizon length, and

*Table 9.* Performance of TimeFuse and base models with increasing input context length $L$. The results ranked **first\*\***/*second\**/*third* are highlighted. Given limited time, we run PatchTST and models newer than it except for TimeMixer, which encounters OOM on V100-32GB GPU when training with lookback length 336 or higher. Generally, longer $L$ benefit PatchTST and PAttn on specific datasets, but not for the rest 3 base models. TimeFuse shows a consistent advantage under various $L$.

| Lookback Length ($L$) | Dataset | TimeFuse | | PatchTST | | TimeXer | | PAttn | | iTransformer | | TimesNet | |
|---|---|---|---|---|---|---|---|---|---|---|---|---|---|
| | | MSE | MAE | MSE | MAE | MSE | MAE | MSE | MAE | MSE | MAE | MSE | MAE |
| | ETTh1 | **0.367\*\*** | **0.391\*\*** | *0.379\** | *0.400\** | 0.382 | 0.403 | 0.390 | 0.405 | 0.447 | 0.444 | 0.389 | 0.412 |
| | ETTh2 | **0.280\*\*** | **0.334\*\*** | 0.292 | 0.345 | *0.285\** | *0.338\** | 0.299 | 0.345 | 0.300 | 0.350 | 0.337 | 0.371 |
| | ETTm1 | **0.306\*\*** | **0.350\*\*** | 0.327 | 0.366 | *0.318\** | *0.356\** | 0.336 | 0.365 | 0.356 | 0.381 | 0.334 | 0.375 |
| $L = 96$ | ETTm2 | **0.170\*\*** | **0.254\*\*** | 0.186 | 0.269 | *0.171\** | *0.256\** | 0.180 | 0.267 | 0.186 | 0.272 | 0.189 | 0.266 |
| | weather | **0.155\*\*** | **0.205\*\*** | 0.172 | 0.213 | *0.157\** | *0.205\** | 0.189 | 0.226 | 0.175 | 0.216 | 0.169 | 0.219 |
| | electricity | **0.133\*\*** | **0.231\*\*** | 0.180 | 0.273 | *0.140\** | 0.242 | 0.196 | 0.276 | 0.149 | *0.241\** | 0.168 | 0.272 |
| | traffic | **0.388\*\*** | **0.255\*\*** | 0.458 | 0.298 | 0.429 | 0.271 | 0.551 | 0.362 | *0.394\** | *0.269\** | 0.590 | 0.314 |
| | Average | **0.257\*\*** | **0.289\*\*** | 0.285 | 0.309 | *0.269\** | *0.296\** | 0.306 | 0.321 | 0.287 | 0.310 | 0.311 | 0.319 |
| | ETTh1 | **0.370\*\*** | **0.392\*\*** | 0.403 | 0.417 | 0.397 | 0.413 | *0.386\** | *0.404\** | 0.435 | 0.441 | 0.438 | 0.450 |
| | ETTh2 | **0.274\*\*** | **0.333\*\*** | *0.286\** | 0.345 | 0.297 | 0.353 | 0.288 | *0.344\** | 0.307 | 0.364 | 0.367 | 0.417 |
| | ETTm1 | **0.279\*\*** | **0.331\*\*** | *0.295\** | 0.350 | 0.306 | 0.356 | 0.298 | *0.345\** | 0.315 | 0.363 | 0.319 | 0.367 |
| $L = 336$ | ETTm2 | **0.155\*\*** | **0.245\*\*** | 0.173 | 0.261 | *0.169\** | 0.261 | 0.170 | *0.258\** | 0.179 | 0.273 | 0.187 | 0.275 |
| | weather | **0.140\*\*** | **0.191\*\*** | *0.154\** | *0.204\** | 0.158 | 0.209 | 0.158 | 0.206 | 0.171 | 0.223 | 0.164 | 0.220 |
| | electricity | **0.130\*\*** | **0.228\*\*** | 0.147 | 0.251 | 0.153 | 0.258 | *0.143\** | *0.239\** | 0.169 | 0.274 | 0.191 | 0.297 |
| | traffic | **0.387\*\*** | **0.267\*\*** | 0.405 | 0.291 | 0.412 | 0.297 | *0.404\** | *0.282\** | 0.442 | 0.330 | 0.605 | 0.345 |
| | Average | **0.248\*\*** | **0.284\*\*** | 0.266 | 0.303 | 0.270 | 0.307 | *0.264\** | *0.297\** | 0.288 | 0.324 | 0.324 | 0.339 |
| | ETTh1 | **0.364\*\*** | **0.393\*\*** | *0.381\** | *0.404\** | 0.389 | 0.413 | 0.381 | 0.407 | 0.435 | 0.444 | 0.434 | 0.450 |
| | ETTh2 | **0.274\*\*** | **0.332\*\*** | 0.290 | 0.348 | *0.286\** | 0.349 | 0.287 | *0.347\** | 0.309 | 0.371 | 0.390 | 0.424 |
| | ETTm1 | **0.282\*\*** | **0.334\*\*** | *0.296\** | 0.350 | 0.307 | 0.358 | 0.301 | *0.347\** | 0.321 | 0.367 | 0.337 | 0.379 |
| $L = 512$ | ETTm2 | **0.157\*\*** | **0.248\*\*** | *0.169\** | 0.260 | 0.171 | 0.260 | 0.171 | *0.260\** | 0.180 | 0.273 | 0.191 | 0.278 |
| | weather | **0.138\*\*** | **0.190\*\*** | *0.151\** | 0.203 | 0.157 | 0.208 | 0.153 | *0.202\** | 0.166 | 0.220 | 0.157 | 0.215 |
| | electricity | **0.127\*\*** | **0.225\*\*** | 0.144 | 0.249 | 0.148 | 0.255 | *0.138\** | *0.236\** | 0.165 | 0.270 | 0.196 | 0.302 |
| | traffic | **0.376\*\*** | **0.262\*\*** | *0.391\** | 0.289 | 0.399 | 0.290 | 0.392 | *0.277\** | 0.434 | 0.325 | 0.597 | 0.329 |
| | Average | **0.245\*\*** | **0.283\*\*** | 0.260 | 0.301 | 0.265 | 0.305 | *0.260\** | *0.296\** | 0.287 | 0.324 | 0.329 | 0.340 |

temporal complexity.

## C. Full Results

Due to space constraints, we report the average performance scores across all prediction lengths for long-term forecasting datasets and short-term forecasting PEMS datasets in the main content. Detailed experimental results are provided in Tables 11 and 12, which show that TIMEFUSE consistently delivers robust performance at various prediction lengths and consistently outperforms the task-specific best individual models across different datasets and metrics.

## D. Discussion on Limitations and Future Directions

While TIMEFUSE demonstrates strong performance across diverse forecasting tasks, several limitations remain that point to promising directions for future work.

**Distribution Shift.** TIMEFUSE relies on meta-training data to learn its sample-wise fusion strategy, and its performance can be influenced by distribution shifts between meta-train (validation) and meta-test (test) sets. We ob-

served that some base models perform strongly on the meta-training set but yield different behavior on the test set, which may affect fusion decisions. In some cases, excluding such models from the model zoo has been observed to improve results. Future work may investigate data augmentation (Lin et al., 2024; He et al., 2025; 2024; Li et al., 2025; Yan et al., 2023a; 2024b;c; Xu et al.; Zheng et al., 2024b; Feng et al., 2022; Jing et al., 2024b; Zeng et al., 2024b; Wei et al., 2025; 2022), adaptation (Yoo et al., 2025a;b; 2024; Zeng et al., 2025; Wu et al., 2024; Bao et al., 2023; Wei et al., 2020), and generation (Jing et al., 2024a; Xu et al., 2024b; Qiu et al., 2024c;b; Wei et al., 2024b;a) techniques to enrich meta-training distributions and support better adaptation under potential distribution shifts (Fu et al., 2022; Bao et al., 2025; Lin et al., 2025; Qiu et al., 2024a; Qiu & Tong, 2024; Tieu et al., 2025; Wei et al., 2021; Wei & He, 2022).

**Fusor Expressiveness.** The current fusor architecture is designed to be simple for general applicability and computational efficiency. As the diversity of tasks increases, exploring more expressive fusor models that can capture complex meta-level patterns becomes an appealing direction. Careful design is needed to balance model complexity and generalization, particularly when meta-training data is

limited.

**Task Contribution Balance.** To address the imbalance in the number of training samples across tasks, we currently adopt a straightforward oversampling strategy. While effective in some settings, this approach may not always capture the optimal balance across tasks with differing sample sizes. Alternative resampling strategies (Liu et al., 2020b;a; 2021; 2024b; Yan et al., 2023b) from the imbalanced classification literature may offer a more flexible and adaptive way to enhance training efficiency and model robustness across tasks.

**Incorporating Spatial or Graph Structure.** Many multivariate time series are associated with spatial structure, such as traffic or weather sensors distributed over geographic regions, where nearby sensors often record similar measurements. These spatial dependencies are often represented using graph structures (Fu et al., 2024a; Ban et al., 2024; Zou et al., 2025) and analyzed using graph mining techniques (Yan et al., 2021b;a; 2022; Li et al., 2023; Lin et al., 2024; Roach et al., 2020; Jing et al., 2022) or learning on graphs (Fu et al., 2024b; Tieu et al., 2024; Zheng et al., 2024a; Qiu et al., 2023; 2022; Xu et al., 2024a; Zheng et al., 2024c; Wang et al., 2025; Jing et al., 2021; 2024c; Zeng et al., 2023b;c; 2024a). Extending the TIMEFUSE framework to explicitly incorporate such spatial or graph-based information could enable more adaptive fusion strategies, particularly for applications where spatial relationships play a central role (Wang et al., 2023; Yan et al., 2024a; Fu et al., 2024c; Fang et al., 2025).

These challenges present opportunities to further enhance the robustness, adaptability, and efficiency of TIMEFUSE in future iterations.

*Table 10.* Performance comparison between TimeFuse and **(i)** advanced ensemble solutions, **(ii)** AutoML ensemble AutoGluon (with '`high_quality`' presets), and **(iii)** pretrained time-series model Chronos-Bolt-Base (finetune/influence with the AutoGluon API). We test the prediction length of 96/24 for long/short-term forecasting due to limited time. The results ranked **first**\*\*/*second*\*/*third* are highlighted.

| Task | Dataset | Metric | Ours TimeFuse | Advanced Ensemble Forward Selection | Advanced Ensemble Portfolio Ensemble | Advanced Ensemble ZeroShot Ensemble | AutoML Ensemble AutoGluon (24 models) | Foundation Model ChronosBolt Finetuned | Foundation Model ChronosBolt Zeroshot |
|---|---|---|---|---|---|---|---|---|---|
| **Long-term Forecasting Multivariate** | ETTh1 | MSE | **0.367**\*\* | *0.380* | 0.381 | *0.373*\* | 0.422 | 0.414 | 0.490 |
| | | MAE | **0.391**\*\* | 0.399 | 0.400 | *0.397*\* | 0.437 | *0.397* | 0.403 |
| | ETTh2 | MSE | **0.280**\*\* | *0.284* | *0.284*\* | 0.298 | 0.309 | 1.405 | 1411793.085 |
| | | MAE | **0.334**\*\* | *0.341*\* | *0.341* | 0.361 | 0.356 | 0.385 | 68.569 |
| | ETTm1 | MSE | **0.306**\*\* | *0.309* | *0.308*\* | 0.328 | 42.945 | 0.930 | 8927.445 |
| | | MAE | **0.350**\*\* | 0.358 | *0.357*\* | 0.372 | 0.492 | 0.488 | 1.505 |
| | ETTm2 | MSE | **0.170**\*\* | *0.173*\* | *0.173* | 0.177 | 3412923.439 | 2.037 | 5908821.446 |
| | | MAE | **0.254**\*\* | *0.261*\* | 0.262 | 0.271 | 219.784 | 1.035 | 289.118 |
| | electricity | MSE | **0.133**\*\* | *0.137* | *0.136*\* | 0.150 | OOT | 0.353 | 13507.261 |
| | | MAE | **0.231**\*\* | 0.240 | *0.240*\* | 0.255 | OOT | 0.280 | 2.041 |
| | weather | MSE | **0.155**\*\* | 0.167 | *0.159*\* | *0.164* | 0.211 | 0.219 | 0.218 |
| | | MAE | **0.205**\*\* | 0.233 | *0.218*\* | 0.219 | 0.263 | 0.258 | 0.256 |
| | traffic | MSE | **0.388**\*\* | *0.405* | *0.397*\* | 0.467 | OOT | 0.904 | 24684.662 |
| | | MAE | **0.255**\*\* | 0.267 | *0.262*\* | 0.299 | OOT | 0.340 | 2.048 |
| | Avg. | MSE | **0.257**\*\* | 0.265 | *0.263*\* | 0.279 | 682593.465 | 11.609 | 1052533.515 |
| | | MAE | **0.289**\*\* | 0.300 | *0.297*\* | 0.311 | 44.266 | 0.598 | 51.992 |
| **Short-term Forecasting Multivariate** | PEMS03 | MAE | **18.551**\*\* | *19.226* | *18.886*\* | 19.669 | 37.392 | 44.713 | 45.925 |
| | | RMSE | **28.966**\*\* | *29.867* | *29.398*\* | 30.700 | 58.613 | 72.054 | 73.835 |
| | | MAPE | **0.158**\*\* | *0.163* | *0.159*\* | 0.164 | 0.235 | 0.306 | 0.384 |
| | PEMS04 | MAE | **22.610**\*\* | *23.643* | *22.900*\* | 24.985 | 33.367 | 32.949 | 37.422 |
| | | RMSE | **35.355**\*\* | *36.889* | *36.098*\* | 38.930 | 50.031 | 50.661 | 55.908 |
| | | MAPE | **0.189**\*\* | 0.207 | *0.192*\* | *0.204* | 0.262 | 0.278 | 0.355 |
| | PEMS07 | MAE | **26.105**\*\* | *27.755* | *27.434*\* | 29.487 | 61.004 | 61.704 | 64.239 |
| | | RMSE | **40.763**\*\* | *42.742* | *42.382*\* | 44.976 | 90.104 | 94.850 | 98.631 |
| | | MAPE | **0.128**\*\* | *0.142* | *0.137*\* | 0.148 | 0.289 | 0.334 | 0.370 |
| | PEMS08 | MAE | **19.125**\*\* | *20.225* | *19.924*\* | 21.150 | 40.538 | 26.423 | 29.556 |
| | | RMSE | **29.748**\*\* | *31.326* | *30.696*\* | 32.709 | 52.422 | 39.718 | 43.971 |
| | | MAPE | **0.123**\*\* | *0.133* | *0.133*\* | 0.137 | 0.216 | 0.168 | 0.199 |
| | Avg. | MAE | **21.598**\*\* | *22.712* | *22.286*\* | 23.823 | 43.075 | 41.447 | 44.285 |
| | | RMSE | **33.708**\*\* | *35.206* | *34.643*\* | 36.829 | 62.792 | 64.321 | 68.086 |
| | | MAPE | **0.149**\*\* | *0.161* | *0.155*\* | 0.163 | 0.250 | 0.271 | 0.327 |
| **Short-term Forecasting Univariate** | NP | MSE | **0.229**\*\* | 0.235 | *0.234*\* | 0.245 | 0.235 | 0.255 | 0.264 |
| | | MAE | **0.260**\*\* | 0.277 | 0.277 | 0.279 | *0.267*\* | *0.275* | 0.279 |
| | PJM | MSE | *0.085*\* | 0.086 | *0.086* | 0.087 | **0.084**\*\* | 0.090 | 0.092 |
| | | MAE | *0.185*\* | 0.191 | 0.189 | 0.191 | **0.182**\*\* | *0.187* | 0.191 |
| | BE | MSE | *0.378*\* | *0.397* | 0.400 | 0.403 | 0.407 | **0.373**\*\* | 0.428 |
| | | MAE | **0.241**\*\* | 0.254 | 0.255 | 0.260 | 0.270 | *0.248*\* | 0.262 |
| | FR | MSE | *0.386* | 0.391 | 0.392 | 0.399 | *0.379*\* | **0.376**\*\* | 0.434 |
| | | MAE | **0.198**\*\* | 0.209 | *0.208* | *0.207*\* | 0.220 | 0.211 | 0.225 |
| | DE | MSE | **0.429**\*\* | 0.455 | *0.451* | 0.452 | *0.437*\* | 0.503 | 0.553 |
| | | MAE | *0.407*\* | 0.438 | 0.435 | 0.434 | **0.389**\*\* | *0.424* | 0.455 |
| | Avg. | MSE | **0.301**\*\* | 0.313 | *0.312* | 0.318 | *0.308*\* | 0.319 | 0.354 |
| | | MAE | **0.258**\*\* | 0.274 | 0.273 | 0.274 | *0.265*\* | *0.269* | 0.282 |

- **OOT: Out-of-time, AutoGluon inference takes over 3 hours.** This is due to several reasons: (i) AutoGluon is for univariate forecasting and must repeatedly predict each of the 321/862 variates on the electricity/traffic dataset. (ii) The predict length significantly influences the inference time of some AutoGluon base models. (iii) A large part of AutoGluon base models cannot be accelerated by GPU.
- **Chronos/AutoGluon abnormally high error on long-term forecast tasks:** We carefully checked the pipeline and confirmed these results. We believe this is because Chronos cannot handle long-term forecasting that extends over its predict length (64) used in pretraining.

*Table 11.* Full long-term forecasting results.

| Forecast MSE | | TFuse (Ours) | TXer (2024) | TMixer (2024) | PAttn (2024) | iTF (2024) | TNet (2023) | PTST (2023) | DLin (2023) | FreTS (2023) | FEDF (2022) | NSTF (2022) | LightTS (2022) | InF (2021) | AutoF (2021) |
|---|---|---|---|---|---|---|---|---|---|---|---|---|---|---|---|
| ETTh1 | 96 | 0.367 | 0.382 | 0.378 | 0.390 | 0.447 | 0.389 | 0.379 | 0.396 | 0.400 | 0.377 | 0.540 | 0.435 | 0.963 | 0.524 |
| | 192 | 0.415 | 0.429 | 0.441 | 0.437 | 0.519 | 0.439 | 0.427 | 0.445 | 0.451 | 0.420 | 0.635 | 0.494 | 1.020 | 0.557 |
| | 336 | 0.462 | 0.467 | 0.500 | 0.498 | 0.554 | 0.494 | 0.478 | 0.487 | 0.509 | 0.458 | 0.781 | 0.549 | 1.030 | 0.555 |
| | 720 | 0.476 | 0.499 | 0.479 | 0.533 | 0.563 | 0.516 | 0.522 | 0.513 | 0.563 | 0.502 | 0.632 | 0.612 | 1.222 | 0.543 |
| | Avg. | 0.430 | 0.444 | 0.450 | 0.464 | 0.521 | 0.460 | 0.452 | 0.460 | 0.481 | 0.439 | 0.647 | 0.522 | 1.059 | 0.545 |
| ETTh2 | 96 | 0.280 | 0.285 | 0.290 | 0.299 | 0.300 | 0.337 | 0.292 | 0.341 | 0.342 | 0.348 | 0.397 | 0.435 | 1.764 | 0.385 |
| | 192 | 0.358 | 0.363 | 0.387 | 0.383 | 0.382 | 0.405 | 0.381 | 0.482 | 0.440 | 0.424 | 0.532 | 0.561 | 2.201 | 0.451 |
| | 336 | 0.404 | 0.412 | 0.427 | 0.423 | 0.424 | 0.459 | 0.418 | 0.593 | 0.538 | 0.457 | 0.569 | 0.683 | 2.173 | 0.490 |
| | 720 | 0.417 | 0.448 | 0.469 | 0.428 | 0.426 | 0.434 | 0.433 | 0.840 | 0.796 | 0.476 | 0.631 | 0.937 | 2.750 | 0.504 |
| | Avg. | 0.365 | 0.377 | 0.393 | 0.383 | 0.383 | 0.409 | 0.381 | 0.564 | 0.529 | 0.426 | 0.532 | 0.654 | 2.222 | 0.457 |
| ETTm1 | 96 | 0.306 | 0.318 | 0.336 | 0.336 | 0.356 | 0.334 | 0.327 | 0.346 | 0.340 | 0.539 | 0.406 | 0.369 | 0.740 | 0.522 |
| | 192 | 0.346 | 0.362 | 0.363 | 0.376 | 0.399 | 0.408 | 0.370 | 0.382 | 0.383 | 0.535 | 0.517 | 0.399 | 0.761 | 0.617 |
| | 336 | 0.380 | 0.395 | 0.390 | 0.407 | 0.433 | 0.413 | 0.398 | 0.415 | 0.420 | 0.681 | 0.583 | 0.430 | 0.872 | 0.640 |
| | 720 | 0.443 | 0.452 | 0.456 | 0.467 | 0.501 | 0.486 | 0.458 | 0.473 | 0.497 | 0.709 | 0.614 | 0.488 | 0.998 | 0.557 |
| | Avg. | 0.369 | 0.382 | 0.386 | 0.397 | 0.422 | 0.410 | 0.388 | 0.404 | 0.410 | 0.616 | 0.530 | 0.421 | 0.843 | 0.584 |
| ETTm2 | 96 | 0.170 | 0.171 | 0.175 | 0.180 | 0.186 | 0.189 | 0.186 | 0.193 | 0.195 | 0.211 | 0.307 | 0.209 | 0.744 | 0.306 |
| | 192 | 0.238 | 0.236 | 0.236 | 0.246 | 0.252 | 0.263 | 0.246 | 0.285 | 0.277 | 0.271 | 0.591 | 0.300 | 0.937 | 0.283 |
| | 336 | 0.295 | 0.295 | 0.297 | 0.308 | 0.315 | 0.320 | 0.310 | 0.385 | 0.370 | 0.328 | 0.497 | 0.393 | 1.278 | 0.330 |
| | 720 | 0.392 | 0.393 | 0.394 | 0.408 | 0.413 | 0.421 | 0.415 | 0.556 | 0.560 | 0.420 | 0.679 | 0.545 | 1.938 | 0.428 |
| | Avg. | 0.274 | 0.274 | 0.275 | 0.286 | 0.292 | 0.298 | 0.289 | 0.355 | 0.350 | 0.308 | 0.518 | 0.362 | 1.224 | 0.337 |
| Weather | 96 | 0.155 | 0.157 | 0.161 | 0.189 | 0.175 | 0.169 | 0.172 | 0.196 | 0.184 | 0.280 | 0.175 | 0.196 | 0.519 | 0.284 |
| | 192 | 0.203 | 0.204 | 0.207 | 0.234 | 0.225 | 0.225 | 0.220 | 0.239 | 0.223 | 0.330 | 0.235 | 0.242 | 0.577 | 0.300 |
| | 336 | 0.260 | 0.261 | 0.263 | 0.287 | 0.280 | 0.281 | 0.279 | 0.281 | 0.272 | 0.328 | 0.335 | 0.290 | 0.971 | 0.353 |
| | 720 | 0.341 | 0.340 | 0.344 | 0.362 | 0.361 | 0.359 | 0.355 | 0.345 | 0.354 | 0.394 | 0.412 | 0.350 | 1.266 | 0.426 |
| | Avg. | 0.240 | 0.241 | 0.244 | 0.268 | 0.260 | 0.259 | 0.257 | 0.265 | 0.258 | 0.333 | 0.289 | 0.269 | 0.834 | 0.341 |
| Electricity | 96 | 0.133 | 0.140 | 0.156 | 0.196 | 0.149 | 0.168 | 0.180 | 0.210 | 0.189 | 0.195 | 0.171 | 0.213 | 0.330 | 0.201 |
| | 192 | 0.151 | 0.157 | 0.170 | 0.197 | 0.164 | 0.187 | 0.187 | 0.210 | 0.192 | 0.202 | 0.184 | 0.222 | 0.357 | 0.221 |
| | 336 | 0.168 | 0.177 | 0.187 | 0.212 | 0.178 | 0.204 | 0.204 | 0.223 | 0.207 | 0.229 | 0.203 | 0.242 | 0.352 | 0.252 |
| | 720 | 0.199 | 0.211 | 0.228 | 0.254 | 0.227 | 0.225 | 0.246 | 0.258 | 0.247 | 0.262 | 0.225 | 0.277 | 0.394 | 0.287 |
| | Avg. | 0.163 | 0.171 | 0.185 | 0.215 | 0.180 | 0.196 | 0.204 | 0.225 | 0.209 | 0.222 | 0.196 | 0.238 | 0.358 | 0.240 |
| Traffic | 96 | 0.388 | 0.429 | 0.474 | 0.551 | 0.394 | 0.590 | 0.458 | 0.697 | 0.565 | 0.654 | 0.615 | 0.769 | 1.056 | 0.673 |
| | 192 | 0.409 | 0.448 | 0.501 | 0.539 | 0.412 | 0.615 | 0.469 | 0.647 | 0.568 | 0.660 | 0.650 | 0.728 | 1.307 | 0.632 |
| | 336 | 0.421 | 0.472 | 0.528 | 0.549 | 0.423 | 0.635 | 0.483 | 0.653 | 0.600 | 0.740 | 0.641 | 0.737 | 1.416 | 0.632 |
| | 720 | 0.457 | 0.515 | 0.548 | 0.581 | 0.458 | 0.656 | 0.517 | 0.694 | 0.661 | 0.792 | 0.658 | 0.785 | 1.480 | 0.673 |
| | Avg. | 0.419 | 0.466 | 0.513 | 0.555 | 0.422 | 0.624 | 0.482 | 0.673 | 0.599 | 0.711 | 0.641 | 0.755 | 1.315 | 0.652 |
| Forecast MAE | | TFuse (Ours) | TXer (2024) | TMixer (2024) | PAttn (2024) | iTF (2024) | TNet (2023) | PTST (2023) | DLin (2023) | FreTS (2023) | FEDF (2022) | NSTF (2022) | LightTS (2022) | InF (2021) | AutoF (2021) |
| ETTh1 | 96 | 0.391 | 0.403 | 0.398 | 0.405 | 0.444 | 0.412 | 0.400 | 0.411 | 0.412 | 0.418 | 0.502 | 0.444 | 0.780 | 0.495 |
| | 192 | 0.420 | 0.435 | 0.430 | 0.439 | 0.483 | 0.442 | 0.432 | 0.440 | 0.443 | 0.444 | 0.565 | 0.478 | 0.794 | 0.517 |
| | 336 | 0.444 | 0.449 | 0.460 | 0.470 | 0.502 | 0.471 | 0.464 | 0.465 | 0.481 | 0.467 | 0.652 | 0.510 | 0.780 | 0.514 |
| | 720 | 0.477 | 0.493 | 0.472 | 0.499 | 0.525 | 0.494 | 0.506 | 0.510 | 0.540 | 0.503 | 0.570 | 0.569 | 0.880 | 0.523 |
| | Avg. | 0.433 | 0.445 | 0.440 | 0.453 | 0.489 | 0.455 | 0.451 | 0.457 | 0.469 | 0.458 | 0.572 | 0.500 | 0.808 | 0.512 |
| ETTh2 | 96 | 0.334 | 0.338 | 0.341 | 0.345 | 0.350 | 0.371 | 0.345 | 0.395 | 0.397 | 0.391 | 0.420 | 0.473 | 1.074 | 0.417 |
| | 192 | 0.384 | 0.389 | 0.401 | 0.395 | 0.400 | 0.415 | 0.404 | 0.479 | 0.449 | 0.437 | 0.487 | 0.538 | 1.202 | 0.453 |
| | 336 | 0.421 | 0.424 | 0.435 | 0.430 | 0.432 | 0.454 | 0.435 | 0.542 | 0.509 | 0.468 | 0.514 | 0.598 | 1.230 | 0.486 |
| | 720 | 0.439 | 0.453 | 0.469 | 0.444 | 0.445 | 0.448 | 0.452 | 0.661 | 0.644 | 0.486 | 0.548 | 0.714 | 1.386 | 0.503 |
| | Avg. | 0.395 | 0.401 | 0.411 | 0.404 | 0.407 | 0.422 | 0.409 | 0.519 | 0.500 | 0.445 | 0.492 | 0.581 | 1.223 | 0.465 |
| ETTm1 | 96 | 0.350 | 0.356 | 0.373 | 0.365 | 0.381 | 0.375 | 0.366 | 0.374 | 0.375 | 0.490 | 0.409 | 0.391 | 0.620 | 0.485 |
| | 192 | 0.372 | 0.383 | 0.384 | 0.383 | 0.402 | 0.414 | 0.390 | 0.391 | 0.398 | 0.494 | 0.457 | 0.406 | 0.627 | 0.529 |
| | 336 | 0.395 | 0.407 | 0.404 | 0.404 | 0.424 | 0.421 | 0.408 | 0.415 | 0.425 | 0.548 | 0.493 | 0.426 | 0.688 | 0.541 |
| | 720 | 0.434 | 0.441 | 0.445 | 0.438 | 0.462 | 0.461 | 0.444 | 0.451 | 0.474 | 0.562 | 0.531 | 0.462 | 0.738 | 0.513 |
| | Avg. | 0.388 | 0.397 | 0.401 | 0.397 | 0.417 | 0.418 | 0.402 | 0.408 | 0.418 | 0.524 | 0.473 | 0.421 | 0.668 | 0.517 |
| ETTm2 | 96 | 0.254 | 0.256 | 0.258 | 0.267 | 0.272 | 0.266 | 0.269 | 0.293 | 0.285 | 0.294 | 0.338 | 0.308 | 0.631 | 0.344 |
| | 192 | 0.297 | 0.299 | 0.298 | 0.308 | 0.312 | 0.312 | 0.306 | 0.361 | 0.351 | 0.329 | 0.467 | 0.375 | 0.746 | 0.341 |
| | 336 | 0.335 | 0.338 | 0.339 | 0.347 | 0.353 | 0.348 | 0.349 | 0.429 | 0.404 | 0.364 | 0.435 | 0.435 | 0.889 | 0.365 |
| | 720 | 0.392 | 0.394 | 0.399 | 0.403 | 0.406 | 0.406 | 0.411 | 0.523 | 0.519 | 0.416 | 0.518 | 0.519 | 1.131 | 0.422 |
| | Avg. | 0.319 | 0.322 | 0.323 | 0.331 | 0.336 | 0.333 | 0.334 | 0.402 | 0.390 | 0.351 | 0.439 | 0.409 | 0.849 | 0.368 |
| Weather | 96 | 0.205 | 0.205 | 0.208 | 0.226 | 0.216 | 0.219 | 0.213 | 0.256 | 0.239 | 0.349 | 0.227 | 0.255 | 0.515 | 0.348 |
| | 192 | 0.247 | 0.247 | 0.251 | 0.264 | 0.258 | 0.264 | 0.256 | 0.299 | 0.274 | 0.384 | 0.277 | 0.299 | 0.539 | 0.361 |
| | 336 | 0.287 | 0.290 | 0.292 | 0.301 | 0.298 | 0.304 | 0.298 | 0.331 | 0.318 | 0.366 | 0.343 | 0.337 | 0.734 | 0.388 |
| | 720 | 0.342 | 0.340 | 0.344 | 0.349 | 0.351 | 0.354 | 0.347 | 0.382 | 0.387 | 0.399 | 0.390 | 0.383 | 0.883 | 0.432 |
| | Avg. | 0.270 | 0.271 | 0.273 | 0.285 | 0.281 | 0.285 | 0.278 | 0.317 | 0.304 | 0.375 | 0.309 | 0.318 | 0.668 | 0.382 |
| Electricity | 96 | 0.231 | 0.242 | 0.248 | 0.276 | 0.241 | 0.272 | 0.273 | 0.302 | 0.277 | 0.309 | 0.272 | 0.316 | 0.415 | 0.318 |
| | 192 | 0.247 | 0.256 | 0.261 | 0.279 | 0.256 | 0.289 | 0.280 | 0.305 | 0.279 | 0.315 | 0.284 | 0.325 | 0.439 | 0.330 |
| | 336 | 0.264 | 0.275 | 0.277 | 0.294 | 0.271 | 0.305 | 0.296 | 0.319 | 0.297 | 0.342 | 0.302 | 0.344 | 0.436 | 0.353 |
| | 720 | 0.296 | 0.306 | 0.313 | 0.327 | 0.311 | 0.323 | 0.328 | 0.350 | 0.333 | 0.365 | 0.324 | 0.371 | 0.457 | 0.382 |
| | Avg. | 0.260 | 0.270 | 0.275 | 0.294 | 0.270 | 0.297 | 0.294 | 0.319 | 0.296 | 0.333 | 0.296 | 0.339 | 0.437 | 0.346 |
| Traffic | 96 | 0.255 | 0.271 | 0.293 | 0.362 | 0.269 | 0.314 | 0.298 | 0.429 | 0.368 | 0.420 | 0.342 | 0.477 | 0.598 | 0.403 |
| | 192 | 0.266 | 0.282 | 0.302 | 0.350 | 0.277 | 0.324 | 0.301 | 0.407 | 0.365 | 0.418 | 0.359 | 0.463 | 0.713 | 0.396 |
| | 336 | 0.274 | 0.289 | 0.308 | 0.352 | 0.283 | 0.338 | 0.307 | 0.410 | 0.374 | 0.457 | 0.353 | 0.464 | 0.788 | 0.396 |
| | 720 | 0.292 | 0.307 | 0.323 | 0.369 | 0.300 | 0.347 | 0.326 | 0.429 | 0.398 | 0.484 | 0.355 | 0.483 | 0.812 | 0.415 |
| | Avg. | 0.272 | 0.287 | 0.306 | 0.358 | 0.282 | 0.331 | 0.308 | 0.419 | 0.376 | 0.445 | 0.352 | 0.472 | 0.728 | 0.402 |

*Table 12.* Full short-term forecasting results.

| Metric MAE | | TFuse (Ours) | TXer (2024) | TMixer (2024) | PAttn (2024) | iTF (2024) | TNet (2023) | PTST (2023) | DLin (2023) | FreTS (2023) | FEDF (2022) | NSTF (2022) | LightTS (2022) | InF (2021) | AutoF (2021) |
|---|---|---|---|---|---|---|---|---|---|---|---|---|---|---|---|
| PEMS03 | 6 | **14.266** | 15.883 | 15.811 | 15.529 | *14.941* | 17.665 | 16.319 | 17.085 | 15.709 | 31.027 | 16.918 | 15.429 | 20.130 | 23.146 |
| | 12 | **15.197** | 17.469 | *15.910* | 18.443 | 16.947 | 18.784 | 19.216 | 20.927 | 18.079 | 33.682 | 23.130 | 17.821 | 20.280 | 35.126 |
| | 24 | **18.551** | *20.349* | 23.130 | 23.694 | 20.372 | 21.599 | 22.962 | 28.073 | 22.113 | 29.629 | 20.963 | 21.169 | 22.115 | 49.254 |
| | Avg. | **16.005** | 17.900 | 18.283 | 19.222 | *17.420* | 19.349 | 19.499 | 22.028 | 18.634 | 31.446 | 20.337 | 18.140 | 20.842 | 35.842 |
| PEMS04 | 6 | **18.936** | 21.105 | 21.373 | 21.092 | 20.128 | 21.695 | 21.726 | 21.818 | 21.011 | 40.668 | 21.227 | *20.073* | 22.521 | 27.033 |
| | 12 | **19.259** | 22.019 | *19.572* | 24.621 | 22.390 | 22.668 | 25.555 | 26.567 | 23.872 | 37.758 | 22.952 | 21.478 | 22.779 | 39.194 |
| | 24 | **22.610** | 24.416 | 29.948 | 32.061 | 27.217 | 24.722 | 32.492 | 34.845 | 29.569 | 33.891 | 24.387 | 25.161 | *24.166* | 59.911 |
| | Avg. | **20.268** | 22.513 | 23.631 | 25.925 | 23.245 | 23.028 | 26.591 | 27.743 | 24.817 | 37.439 | 22.855 | *22.237* | 23.155 | 42.046 |
| PEMS07 | 6 | **21.205** | 26.042 | 23.542 | 23.228 | *22.243* | 26.487 | 25.604 | 25.027 | 23.077 | 56.636 | 28.244 | 22.694 | 29.543 | 48.333 |
| | 12 | **21.654** | 25.627 | *22.127* | 27.409 | 25.061 | 27.380 | 28.721 | 31.468 | 27.044 | 57.879 | 28.580 | 25.365 | 29.789 | 63.533 |
| | 24 | **26.105** | *28.416* | 33.799 | 36.199 | 30.498 | 30.064 | 37.596 | 43.600 | 34.793 | 46.379 | 30.111 | 29.955 | 31.301 | 76.314 |
| | Avg. | **22.988** | 26.695 | 26.489 | 28.946 | *25.934* | 27.977 | 30.641 | 33.365 | 28.304 | 53.631 | 28.978 | 26.005 | 30.211 | 62.726 |
| PEMS08 | 6 | **14.930** | 16.740 | 16.637 | 16.456 | *15.661* | 18.946 | 16.639 | 18.028 | 16.439 | 40.313 | 17.693 | 15.751 | 20.979 | 26.978 |
| | 12 | **15.348** | 18.163 | *15.687* | 18.872 | 17.215 | 19.943 | 20.023 | 21.191 | 18.378 | 36.794 | 19.814 | 17.467 | 22.934 | 36.776 |
| | 24 | **19.125** | *20.937* | 25.019 | 24.430 | 21.057 | 22.213 | 25.207 | 28.321 | 22.699 | 29.598 | 21.629 | 21.090 | 24.698 | 44.481 |
| | Avg. | **16.468** | 18.613 | 19.114 | 19.920 | *17.978* | 20.367 | 20.623 | 22.513 | 19.172 | 35.569 | 19.712 | 18.103 | 22.870 | 36.078 |

| Metric RMSE | | TFuse (Ours) | TXer (2024) | TMixer (2024) | PAttn (2024) | iTF (2024) | TNet (2023) | PTST (2023) | FreTS (2023) | FEDF (2022) | NSTF (2022) | LightTS (2022) | DLin (2022) | InF (2021) | AutoF (2021) |
|---|---|---|---|---|---|---|---|---|---|---|---|---|---|---|---|
| PEMS03 | 6 | **22.235** | 23.876 | 24.497 | 24.313 | *23.314* | 28.235 | 25.664 | 27.315 | 24.730 | 44.167 | 26.435 | 23.861 | 31.024 | 33.683 |
| | 12 | **23.711** | 26.522 | *24.578* | 29.237 | 26.724 | 29.713 | 29.729 | 33.507 | 28.502 | 48.111 | 35.845 | 27.169 | 31.499 | 50.224 |
| | 24 | **28.966** | *31.220* | 36.229 | 37.774 | 32.117 | 34.198 | 36.772 | 45.149 | 35.078 | 42.545 | 32.752 | 31.840 | 34.672 | 74.000 |
| | Avg. | **24.971** | *27.206* | 28.435 | 30.442 | 27.385 | 30.715 | 30.722 | 35.324 | 29.437 | 44.941 | 31.677 | 27.624 | 32.398 | 52.636 |
| PEMS04 | 6 | **30.338** | *31.934* | 33.606 | 33.612 | 32.180 | 34.250 | 33.790 | 34.495 | 33.210 | 58.083 | 33.788 | 32.030 | 36.552 | 40.349 |
| | 12 | **30.929** | 33.554 | *31.238* | 38.856 | 35.739 | 35.574 | 39.076 | 40.904 | 37.265 | 53.603 | 36.031 | 33.865 | 36.780 | 55.733 |
| | 24 | **35.355** | *37.109* | 45.923 | 50.108 | 42.821 | 38.403 | 49.747 | 52.904 | 45.463 | 48.733 | 38.099 | 38.212 | 38.422 | 82.180 |
| | Avg. | **32.207** | *34.199* | 36.922 | 40.859 | 36.913 | 36.076 | 40.871 | 42.768 | 38.646 | 53.473 | 35.973 | 34.702 | 37.252 | 59.421 |
| PEMS07 | 6 | **33.432** | 37.931 | 36.471 | 36.319 | *34.950* | 41.903 | 37.786 | 38.507 | 35.841 | 76.475 | 43.350 | 35.161 | 47.256 | 64.889 |
| | 12 | **34.528** | 38.109 | *35.089* | 42.679 | 39.564 | 43.602 | 43.247 | 47.167 | 41.573 | 78.103 | 44.804 | 38.917 | 47.676 | 83.471 |
| | 24 | **40.763** | *42.496* | 52.107 | 55.990 | 48.301 | 47.763 | 56.559 | 64.849 | 52.731 | 64.466 | 47.829 | 45.295 | 49.086 | 104.071 |
| | Avg. | **36.241** | *39.512* | 41.222 | 44.996 | 40.938 | 44.423 | 45.864 | 50.174 | 43.382 | 73.015 | 45.328 | 39.791 | 48.006 | 84.144 |
| PEMS08 | 6 | **23.570** | 24.859 | 25.930 | 26.133 | 24.869 | 29.863 | 26.181 | 28.614 | 26.140 | 56.395 | 27.592 | *24.577* | 32.029 | 38.493 |
| | 12 | **24.328** | 27.116 | *24.738* | 29.882 | 27.467 | 31.215 | 30.742 | 32.961 | 29.110 | 52.460 | 30.628 | 27.043 | 35.299 | 50.732 |
| | 24 | **29.748** | *31.906* | 38.135 | 38.467 | 33.646 | 34.530 | 38.916 | 43.542 | 35.498 | 42.456 | 33.707 | 32.004 | 37.927 | 62.418 |
| | Avg. | **25.882** | 27.960 | 29.601 | 31.494 | 28.661 | 31.869 | 31.946 | 35.039 | 30.249 | 50.437 | 30.643 | *27.875* | 35.085 | 50.548 |

| Metric MAPE | | TFuse (Ours) | TXer (2024) | TMixer (2024) | PAttn (2024) | iTF (2024) | TNet (2023) | PTST (2023) | FreTS (2023) | FEDF (2022) | NSTF (2022) | LightTS (2022) | DLin (2022) | InF (2021) | AutoF (2021) |
|---|---|---|---|---|---|---|---|---|---|---|---|---|---|---|---|
| PEMS03 | 6 | **0.118** | 0.153 | 0.139 | 0.126 | *0.122* | 0.151 | 0.134 | 0.144 | 0.127 | 0.296 | 0.145 | 0.132 | 0.176 | 0.230 |
| | 12 | **0.130** | 0.165 | *0.142* | 0.150 | 0.142 | 0.162 | 0.169 | 0.187 | 0.147 | 0.307 | 0.203 | 0.164 | 0.175 | 0.316 |
| | 24 | **0.158** | 0.193 | 0.197 | 0.195 | *0.173* | 0.191 | 0.190 | 0.256 | 0.184 | 0.287 | 0.189 | 0.208 | 0.187 | 0.477 |
| | Avg. | **0.135** | 0.170 | 0.159 | 0.157 | *0.146* | 0.168 | 0.164 | 0.196 | 0.153 | 0.297 | 0.179 | 0.168 | 0.179 | 0.341 |
| PEMS04 | 6 | **0.154** | 0.201 | 0.186 | 0.171 | 0.169 | 0.183 | 0.181 | 0.195 | 0.174 | 0.323 | 0.180 | *0.168* | 0.189 | 0.243 |
| | 12 | **0.160** | 0.206 | *0.169* | 0.192 | 0.181 | 0.193 | 0.220 | 0.248 | 0.197 | 0.324 | 0.197 | 0.181 | 0.193 | 0.332 |
| | 24 | **0.189** | 0.223 | 0.247 | 0.253 | 0.221 | 0.215 | 0.270 | 0.316 | 0.241 | 0.302 | 0.209 | 0.232 | *0.202* | 0.478 |
| | Avg. | **0.167** | 0.210 | 0.201 | 0.205 | *0.190* | 0.197 | 0.224 | 0.253 | 0.204 | 0.316 | 0.195 | 0.194 | 0.195 | 0.351 |
| PEMS07 | 6 | **0.099** | 0.135 | 0.113 | 0.109 | *0.109* | 0.130 | 0.130 | 0.121 | 0.112 | 0.282 | 0.143 | 0.110 | 0.144 | 0.279 |
| | 12 | **0.104** | 0.138 | *0.108* | 0.129 | 0.122 | 0.134 | 0.144 | 0.169 | 0.132 | 0.305 | 0.140 | 0.128 | 0.145 | 0.373 |
| | 24 | **0.128** | 0.154 | 0.166 | 0.175 | *0.146* | 0.150 | 0.197 | 0.238 | 0.172 | 0.247 | 0.151 | 0.154 | 0.154 | 0.391 |
| | Avg. | **0.110** | 0.142 | 0.129 | 0.138 | *0.126* | 0.138 | 0.157 | 0.176 | 0.139 | 0.278 | 0.145 | 0.131 | 0.148 | 0.348 |
| PEMS08 | 6 | **0.090** | 0.112 | 0.104 | 0.097 | *0.095* | 0.120 | 0.101 | 0.109 | 0.098 | 0.243 | 0.112 | 0.098 | 0.139 | 0.181 |
| | 12 | **0.097** | 0.124 | *0.102* | 0.115 | 0.106 | 0.129 | 0.132 | 0.135 | 0.113 | 0.225 | 0.127 | 0.113 | 0.150 | 0.254 |
| | 24 | **0.123** | 0.142 | 0.170 | 0.150 | *0.132* | 0.146 | 0.162 | 0.186 | 0.143 | 0.202 | 0.145 | 0.140 | 0.161 | 0.301 |
| | Avg. | **0.103** | 0.126 | 0.125 | 0.121 | *0.111* | 0.131 | 0.132 | 0.143 | 0.118 | 0.223 | 0.128 | 0.117 | 0.150 | 0.246 |

