# OpenReview forum: "Breaking Silos: Adaptive Model Fusion Unlocks Better Time Series Forecasting"
_ICML.cc/2025/Conference — ICML 2025 poster_

### Official Review · Reviewer_oty5 · 2025-02-18

**Overall Recommendation:** 2

**Summary:**

The paper proposes an approach to learn the weights of an ensemble of forecasting models based on meta-features of datasets. The approach first featurizes a time-series dataset then predicts optimal weights of models and average them to obtain predictions. The model is trained with a collection of datasets to learn the optimal weight combinations conditioned on the meta-features to a dataset. Experiments are conducted on real-world datasets against time-series base models.

## update after rebuttal

Based on the rebuttal, I decided to increase my score. I still think the paper will need another submission as the changes will be consequential.

**Results comparison.** I reviewed the results provided by the author and think they will greatly improve the paper, they provide much better baselines than the initial ones.

However, I cannot provide a full assessment as those results would need a major revision of the paper and another full round of review to carefully verify the fully redacted experimental protocol. For instance, how the ensemble are selected (validation error), why AG / Chronos provides large numbers (could be that the method fail catastrophically but it needs some justification given the high scores in public leaderboards), …

>    In contrast, TimeFuse can adaptively predict the optimal weights for each test sample.

I totally see that this could explain the performance improvement, the only caveat is that I think the evaluation setup is a bit restricted, see my comment bellow on the evaluation protocol.

**Evaluation protocol.** I appreciate the clarification on zero-shot results: you perform inference on a dataset that was not seen during training which is what I understood when reading your paper.

My point is that most experiments are not done in this setup (only Tab 5 does it) and even in this setup, lots of datasets are highly similar (eg training on PEMS04/07/08 and evaluating on PEMS03).

However, other methods such as Chronos Bolt are really in this setup where they are evaluated on time series that were not seen. I believe the experimental protocol will carry more weight if most of the pretraining was done on a set of datasets and evaluation was performed on a distinct one (for instance say using gift-eval training split and evaluating on their test datasets).

Based on the rebuttal, I decided to increase my score. I still think the paper will need another submission as the changes will be consequential.

**Claims And Evidence:**

Most of your results are reported on the datasets used for training your meta-learning model. However, the standard practice is to report on hold-out datasets as the performance can be inflated otherwise (see [Auto-Sklearn 2.0: Hands-free AutoML via Meta-Learning](https://www.jmlr.org/papers/v23/21-0992.html) for instance). If you would report performance on those datasets, then you should compare with methods that are also trained on those datasets, for instance some foundational models such as Chronos-bold fine-tuned on those datasets and the performance will be likely quite better. In Tab 5, you report performance on “unseen” datasets but those are mostly the same as it is just the prediction length that is being changed, you just have PEMS which is a true old out dataset.

**Essential References Not Discussed:**

The paper seems to claim that it is the first to consider ensemble across different models/architectures:

> these studies are confined to a single-model architecture, focusing on training multiple same-type models based on different dataset views for a static homogeneous ensemble. Our work, in contrast enables dynamic heterogeneous ensemble of models with various architectures

At least, it provides no previous reference whereas considering ensemble across models is an obvious approach and has been discussed for instance in those two papers: [AutoGluon-TimeSeries: AutoML for Probabilistic Time Series Forecasting](https://arxiv.org/abs/2308.05566) and [Multi-Objective Model Selection for Time Series Forecasting](https://arxiv.org/pdf/2202.08485) to mention just two (there are also many references outside of time-series, for instance AutoGluon-Tabular or AutoSklearn2 that I mentioned above). I would also recommend referring to [hasson23a](https://proceedings.mlr.press/v202/hasson23a/hasson23a.pdf) for a discussion on ensembling where the conditioning is done on items, timestamps and forecasting horizon.

**Experimental Designs Or Analyses:**

Yes, they currently suffer from two limitations:
* the performances are mostly reported on the datasets used for training as opposed to what is frequently being done in meta-learning
* only very naive baselines are considered for the ensembling

**Methods And Evaluation Criteria:**

In addition to discuss more recent prior work on ensembling done for time-series, the paper should also consider non naive-ensembling baselines. Taking only the mean and median in particular is a really weak baseline and it is the only one considered. At the very least, I would recommend to compare with Caruana approach used in AutoGluon-timeseries and AutoSklearn2. I would also recommend to look at portfolio/zeroshot configurations as a second non-naive baseline (described here for instance [Auto-Sklearn 2.0: Hands-free AutoML via Meta-Learning](https://www.jmlr.org/papers/v23/21-0992.html)).

**Other Comments Or Suggestions:**

NA

**Other Strengths And Weaknesses:**

Strength:
* The paper is well written and easy to read.
* The approach proposed is sound and conditioning on meta-features could improve the final performance of ensembles

Weaknesses:
* Very weak baselines considered
* Performance is mostly reported in the training (meta) dataset
* Discussion in related work of ensemble for time-series forecasting mostly missing

**Questions For Authors:**

I would recommend to perform all evaluations on unseen datasets using leave-one-out between one of the 16 datasets and also to compare with non-naive ensembling methods such as Caruana or portfolio configurations.

**Relation To Broader Scientific Literature:**

The forecasting part is well addressed but the part discussing ensembling is very limited and does not include recent work on ensembling for time-series.

**Theoretical Claims:**

NA

---

> ### Author Rebuttal · Authors · 2025-03-31
>
> **Thank you for the thoughtful review and constructive feedback! With our best efforts, we conducted numerous additional analyses to address your concerns. The full results are at https://anonymous.4open.science/api/repo/ICML25Reb-034C/file/Results.pdf?v=b00206ad. Please understand that due to the strict 5000-character limit, we can only summarize the most critical results and insights in the rebuttal :)**
>
> # Q1: Compare with AutoGluon/AutoSklearn2 ensemble and Chronos-Bolt
>
> We implement and test the following approaches following your suggestions:
>
> - **Ensemble**: forward selection (Caruana), portfolio, and zero-shot techniques from  AutoGluon and AutoSklearn2.
> - **Foundation Model**: Chronos-Bolt-Base, both zero-shot and finetuned.
> - **AutoML Ensemble**: AutoGluon-timeseries with all 24 available base models (with `high_quality` presets).
>
> They are trained and evaluated on all 16 datasets, which are categorized into 3 task types:
> - **Long-term Multivariate**: ETTh1/h2/m1/m2, Weather, Electricity, Traffic
> - **Short-term Multivariate**: PEMS03/04/07/08
> - **Short-term Univariate**: EPF (NP/PJM/BE/FR/DE)
>
> We report the averaged MAE for each task type below (with **1st***/**2nd**/*3rd* best highlighted), please see full results for each dataset and metrics in **Table 1** in the link.
>
> |Task Type|TimeFuse|ForwardEns|PortfolioEns|ZeroShotEns|AutoGluon|ChronosBoltFT|ChronosBolt|
> |-|-|-|-|-|-|-|-|
> Long-Multi|**0.289***|*0.300*|**0.297**|0.311|44.266|0.598|51.992|
> Short-Multi|**21.598***|*22.712*|**22.286**|23.823|43.075|41.447|44.285|
> Short-Uni|**0.258***|0.274|0.273|0.274|**0.265**|*0.269*|0.282|
>
> ## Key Findings
>
> 1. **TimeFuse shows a clear advantage across various task types, and we summarize the reason below.**
> 2. **Caruana/portfolio/zero-shot ensemble search the optimal fusion weights at dataset-level, but still statistic at test time.** In contrast, TimeFuse can adaptively predict the optimal weights for each test sample.
> 3. **Chronos is limited to univariate forecasting only and is pretrained with a short predict length (64)**. In Chronos's paper, evaluations are also limited to univariate short-term (4-56) tasks. Our experiments reveal that Chronos suffers in long-term OR multivariate forecasting tasks. While fine-tuning offers improvements, its performance on such tasks is still far from optimal.
> 4. **AutoGluon shares similar limitations with Chronos as it also only supports univariate prediction, and Chronos is one of its main base models.** Additionally, AutoGluon inference is notably slow on long-multi tasks (e.g., needs ~20 hours for predict on traffic dataset with 862 variates), primarily due to (i) the need for independent inference per variate, and (ii) many AutoGluon base models lacking GPU acceleration and running solely on CPUs.
>
> # Q2: Clarification on the evaluation protocol
>
> We appreciate your point on the necessity of testing meta-learning methods on unseen datasets, but we would like to clarify two aspects:
>
> 1. **Table 5 presents valid zero-shot results**: TimeFuse does not train on data from the target dataset in any predict length for zero-shot results. For instance, the results for ETTh1 are obtained with the fusor trained solely on 6 other long-term forecasting datasets. Similarly, results for PEMS03 involve training the fusor exclusively with data from PEMS04/07/08.
> 2. **TimeFuse is not a "pure" meta-learning approach**: We note that TimeFuse is a model fusion framework, it cannot directly predict new datasets without training base models on the new data first.
>
> **Therefore, we opted to compare TimeFuse with other methods (e.g., Caruana/portfolio/Chronos) that also access new data for fine-tuning/searching optimal weights to answer Q1, following your suggestion.**
>
> # Q3: Discussion of related works
>
> We greatly appreciate the suggested papers. We have read these articles and the references cited therein, gaining significant insights from them. We will revise our paper to more accurately reflect our position. Specifically, we believe that compared with existing works, **TimeFuse's core novelty and contribution lie in introducing a methodology for instance-level dynamic ensembling based on the input sample's meta-features, and demonstrating its broad effectiveness in practical applications**. We will discuss these additional related works in the paper to more accurately and comprehensively articulate TimeFuse's relationship with existing research and its position within the literature.
>
> **Finally, we want to thank you again for the thoughtful review, they have been very helpful in improving this paper. We’ve dedicated over 60 hours to implementing the new algorithms, training models, testing, and organizing results, and we will continue expanding those results to further enhance the paper’s quality. We hope these responses address your concerns and would be glad to continue the discussion if you have any further questions!**

---

### Official Review · Reviewer_VM6w · 2025-03-08

**Overall Recommendation:** 3

**Summary:**

This paper introduces TIMEFUSE, a novel framework designed for adaptive fusion of multiple heterogeneous forecasting models to enhance time series forecasting. Key findings indicate that no single model consistently outperforms others across all samples; each excels in specific scenarios. The method addresses this by adaptively fusing model outputs based on input meta-features (statistical, temporal, spectral). A learnable fusor dynamically determines optimal weights for each model. Extensive experiments show that TIMEFUSE outperforms individual state-of-the-art models, achieving improved accuracy on up to 95.1% of test samples.

**Claims And Evidence:**

Yes, the authors’ claims: that (1) no universal model winner and (2) the effectiveness of adaptive fusion have been validated by empirical results.

**Essential References Not Discussed:**

Yes, the author missed discussing a related work. Although the following work didn't employ that many base models and extract different meta-features, the overall problem it frames is very similar:

Han et al., (ICDM 2022), Dynamic Combination of Heterogeneous Models for Hierarchical Time Series

**Experimental Designs Or Analyses:**

The experimental designs and analyses are mostly sound. The choice of baseline models includes a diverse set of recent state-of-the-art models, and the datasets used are standard benchmarks widely recognized in forecasting research.

However, two things are not very clear:

(1) It seems that the fusor is trained on a hold-out validation set and directly minimizes the overall forecasting loss; then what is the training objective of the individual base model? Do they minimize the overall forecasting loss or its own specific loss?

(2) The authors only compare the proposed approach with static ensemble methods and single base models, while there could probably be many other dynamic ensemble methods that produce (soft) combination of models, or MoE that combine predictions from multiple experts. The authors should have a discussion on this aspect.

**Methods And Evaluation Criteria:**

The proposed methods and evaluation criteria presented in the paper are well-suited for addressing the problem at hand. Given the key insight that no single forecasting model excels universally, it is natural to combine the strengths of multiple base forecasting models to achieve consistently strong performance across all tasks. Moreover, extracting multiple meta-features is a reasonable approach that standardizes the input for each forecasting model without requiring separate processing to meet each model’s specific requirements.

**Other Comments Or Suggestions:**

None

**Other Strengths And Weaknesses:**

Weakness:

1. The framework requires specifying a fixed set of predictors. If there are new available models one needs to retrain the fuser to obtain the updated set of weights.

2. In line 234, oversampling all datasets to match the size of the largest task may break the internal structure of that dataset; particularly for time series with strong temporal dependency.

**Questions For Authors:**

1. Do all forecasting base models accept the same input format? Should there be an extra processing step to obtain model-specific input?

2. It is still not clear to me why training on raw features will lead to overfitting. The number of samples that raw features provide should be much more than aggregated values from meta-features.

**Relation To Broader Scientific Literature:**

Unlike traditional static ensemble methods (Kourentzes et al., 2014; Oliveira & Torgo, 2015), TIMEFUSE dynamically integrates diverse models using interpretable meta-features inspired by prior research on feature extraction (Barandas et al., 2020). It extends earlier ensemble ideas (Yu et al., 2017; Choi & Lee, 2018) by enabling heterogeneous models and dynamic sample-level weighting, thus bridging the gap between ensemble methods and state-of-the-art single-model forecasting techniques (Wang et al., 2024; Wu et al., 2023).

**Theoretical Claims:**

There are no theoretical claims.

---

> ### Author Rebuttal · Authors · 2025-03-31
>
> **Thank you for your recognition and thoughtful review! We've conducted extensive analyses to address your concerns. Full results: https://anonymous.4open.science/api/repo/ICML25Reb-034C/file/Results.pdf?v=b00206ad. Please understand that due to the 5000-character limit, we can only summarize key findings here.**
>
> # E1: Training objective of base models
>
> **Base models are trained using their standard prediction losses; the overall (fused) forecasting loss in Eq (1) and (2) is only for training the fusor.** In other words, the base models’ training and inference remain entirely standard—we do not modify their input format, loss functions, or inference procedures. Please refer to Appendix A.3 Implementation Details – Base Forecasting Models for more details.
>
> # E2: Compare with advanced ensemble/MoE algorithms
>
> **Thank you for the great suggestion. We test 4 more advanced ensemble methods from AutoSklearn2 and AutoGluon:**
>
> 1. **Forward Selection**: Builds an ensemble by iteratively adding the model that most improves current predictions on a validation set.
> 2. **Portfolio**: Optimizes the model weights to directly minimize validation loss, similar to risk minimization.
> 3. **Zero-Shot**: Computes the agreement/similarity among model predictions and assigns higher weights to models that have more agreement with others.
> 4. **AutoGluon**: (AutoML) learns the optimal ensemble of 24 base models (with `high_quality` presets).
>
> They are trained and evaluated on all 16 datasets:
> - **Long-term Multivariate (LM)**: ETTh1/h2/m1/m2, Weather, Electricity, Traffic
> - **Short-term Multivariate (SM)**: PEMS03/04/07/08
> - **Short-term Univariate (SU)**: EPF (NP/PJM/BE/FR/DE)
>
> We report the averaged MAE for each task type below (with **1st***/**2nd** best highlighted), please see full results for each dataset and metrics in **Table 1** in the link.
>
> |Task Type|TimeFuse|ForwardEns|PortfolioEns|ZeroShotEns|AutoGluon|
> |-|-|-|-|-|-|
> LM|**0.289***|*0.300*|**0.297**|0.311|44.266|
> SM|**21.598***|*22.712*|**22.286**|23.823|43.075|
> SU|**0.258***|0.274|0.273|0.274|**0.265**|*
>
> ## Key Findings
>
> 1. **TimeFuse consistently outperforms others across diverse task types.**
> 2. **Caruana, portfolio, and zero-shot ensembles optimize fusion weights at the dataset level, but still statistic at test time and thus underperform TimeFuse's sample-level adaptive ensemble.**
> 3. **AutoGluon is limited to univariate forecasting and is extremely slow for long-multivariate tasks, due to per-variate inference and lack of GPU support in many base models.**
>
> We note that we also evaluated the pretrained foundation time-series model Chronos. Due to space constraints, please kindly refer to our response to **Reviewer oty5 Q1** for more details.
>
> # R1: Related work
>
> **Thank you for pointing out this related work. We have carefully reviewed the paper and its references and will include a detailed discussion in our revised version.**
>
> # W1 & W2: Clarification on expanding model zoo and oversampling
>
> We appreciate the weaknesses you pointed out and would like to clarify two points:
>
> 1. **Retraining the fusor incurs minimal computational cost**—usually just a few minutes. This allows easy integration of new models by simply adding their predictions to the meta-training data and retraining the fuser.
> 2. **We use oversampling to balance the data distribution and prevent the pattern of minority datasets from being overwhelmed by larger ones.** We understand concerns about its impact on data representation quality, a potential solution is to adopt more advanced data augmentation techniques to achieve distributional balance.
>
> # Q1: Input format of base models
>
> **Yes, all forecasting base models accept the same input format.** As mentioned in our response to E1, the base models’ training and inference remain entirely standard, with no additional processing required.
>
> # Q2: Why not raw features?
>
> **Sorry for the confusion. We’d like to clarify that using raw features directly as meta features results in higher-dimensional meta features, rather than more samples.** For example, consider a input time series sample $X$ with $D$ variates and length $L$. Using raw features means treating $X\in R^{(L,D)}$ as a single input sample (instead of $D$ samples) to the fusor. In contrast, meta features compress information from $X$ into a lower-dimensional space, reducing the risk of overfitting when learning from complex, high-dimensional raw features. We also ran additional experiments using raw features to validate this point, please see **Table 6** in the full results. **Results show that using meta features consistently outperforms raw features by a significant margin across all datasets.**
>
> **Thank you again for your recognition :) We’ve devoted over 60 hours to get the new results and analysis, which we believe significantly strengthen the paper. We hope our responses have addressed your concerns and would be glad to discuss any further questions.**

---

> > ### Comment · Reviewer_VM6w · 2025-04-02
> >
> > I have read the rebuttal and my concern was mostly addressed. I also thank the authors for additional efforts on new experiments to strengthen the paper. Overall the paper is technically novel and therefore I maintain my score.

---

> > > ### Author Response · Authors · 2025-04-03
> > >
> > > **Dear Reviewer VM6w,**
> > >
> > > **Again, we sincerely appreciate your thoughtful review and encouraging feedback!**
> > >
> > > We are pleased to hear that your previous concerns are mostly addressed. Should you have any additional suggestions, we would be more than happy to engage in further discussions and make any necessary refinements to the manuscript :)
> > >
> > > **All the best wishes,**
> > > **Authors**

---

### Official Review · Reviewer_tYpH · 2025-03-10

**Overall Recommendation:** 3

**Summary:**

The manuscript introduces TimeFuse, a fusion model for time series fusion model. Specifically, TimeFuse uses the outputs of a model zoo and uses meta features of input time series to train a fusion model that predicts the ensemble weights of the individual models from the model zoo. The meta feature set uses various feature sets (statistical, temporal, spectral, multivariate). The authors evaluate their method on 16 datasets and compare the results of TimeFuse with simple mean/median ensemble strategies, along with other further analysis (size of the model zoo, ablation of the meta features, zero-shot performance).

**Claims And Evidence:**

One major claim of the paper is that TimeFuse as an ensemble method outperforms individual models and other ensemble models (top-k mean/median). Another major claim of the paper is that this can be effectively achieved by the presented meta-feature set. Overall, the evidence presented in the paper is through empirical evaluation on the evaluated datasets that beyond mere error reduction provides insight into the fusion weights, size of the model zoo, and learned fusor weights. While the overall evaluation is sound, there are some specific shortcomings of the evaluation that I will address in the next section.

**Essential References Not Discussed:**

As mentioned throughout the review, these essential references are missing either in discussion and/or evaluation

AutoGluon-TimeSeries: (O. Shchur, A. C. Turkmen, N. Erickson, H. Shen, A. Shirkov, T. Hu, and B. Wang. AutoGluon-Timeseries: Automl for probabilistic time series forecasting. In International Conference on Automated Machine Learning, pages 9–1. PMLR, 2023.)

Caruana, R., Niculescu-Mizil, A., Crew, G., and Ksikes, A. (2004). Ensemble selection from libraries of models. In Proceedings of the twenty-first international conference on Machine learning, page 18.

Pretrained time series models should also be discussed as they would be an alternative way to obtain zero-shot results.

**Experimental Designs Or Analyses:**

One specific issue with the specific setup in this paper is that the historical window length is fixed to 96 time steps. One issue that this is introduces is that some of the baseline methods perform much worse in the benchmark presented in this work compared to their original paper. For example, I compared the results in Table 2 in this work with the original PatchTST paper (and with this recent position paper: https://arxiv.org/abs/2502.14045) and found that the error for PatchTST in this work is consistently higher. This is likely attributed to the 96 historical window length As such, the improvement that is showed in this paper is only true relative to some baselines that have been run under suboptimal conditions (96 instead of 512 historical window length). This likely also affects the baseline results TSMixer. I appreciate that there has been earlier work that indicated that the performance of some of the transformer variants does not improve with longer input length. But I think the suboptimal conditions chosen here for PatchTST and TSMixer impact the findings. I would suggest the authors to compare against the baseline models when run against longer input windows.

**Methods And Evaluation Criteria:**

### Datasets

The authors use the widely used long-term forecasting benchmark along with the PEMS and EPF benchmark. Since the evaluation covers several benchmarks, I think it the manuscript offers a sound empirical evaluation of their method. However, the long-term time series benchmark has limitations. These have been explored in recent position papers (https://arxiv.org/abs/2502.14045) and in the NeurIPS 2024 time series workshop (https://neurips.cc/virtual/2024/workshop/84712#wse-detail-108471, see Christoph Bergmeir's talk). I think it is important for the time series forecasting filed to move towards more extensive evaluation which is proposed in several recent papers (Gifteval: https://arxiv.org/abs/2410.10393; FEV (from the Chronos paper): https://openreview.net/forum?id=gerNCVqqtR).

In addition, the evaluation presented here has specific issues that I further elaborate in the "Experimental Designs Or Analyses" section.

### Baselines
The authors use the individual models in the model zoo as baselines along with mean/median top-k ensembles. While these baselines are sound, I think additional baselines need to be added to effectively support the claims of this work.

1/ Table 6 shows the ablation and comparison to TSFEL features. Since one of the contributions of this paper is the meta feature set, I think the TSFEL results should be added to the main results section with all evaluated forecasting horizons. The improvements by the introduced meta feature set are small, so I'm wondering if the improvement holds up for the entire benchmark.

2/ I think the mean/median ensembles are too simplistic to serve as a baseline for other ensemble approaches and I also think the the comparison to other ensemble-based approaches is missing. I would suggest to compare their method against AutoGluon-TimeSeries as an alternative AutoML package that implements an ensemble/meta method for time series forecasting. I appreciate that the model zoo might be rather different, but it would give a meaningful evaluation against other established AutoML/ensemble packages in that domain.

 Additionally, I would suggest that the authors evaluate their method against the ensemble method implement in AutoGluon (https://openreview.net/forum?id=XHIY3cQ8Tew) with their model zoo (forward selection algorithm)

Caruana, R., Niculescu-Mizil, A., Crew, G., and Ksikes, A. (2004). Ensemble selection from libraries of models. In Proceedings of the twenty-first international conference on Machine learning, page 18.

These comparison would substantiate the claims in the paper and demonstrate improvements against stronger baselines/ensembling methods.

3/ The authors mention that raw features would not work as well as the proposed ensemble approach. However, I would kindly ask the authors to substantiate the claim by showing benchmark results with raw features.

**Other Comments Or Suggestions:**

n/a

**Other Strengths And Weaknesses:**

The paper is clearly written and the empirical study is presented well. I appreciate the simplicity of the approach and I think that the work here gives good suggestions the training methodology through oversampling and alternating batching. As discussed, the weaknesses are mostly with specific issues in the experimental setup and choice of baselines.

**Questions For Authors:**

I summarize my main questions to the authors here:

1/ How would the individual model results change if the input context length is increased to 336 or 512 as proposed by earlier work (for example PatchTST)? Would TimeFuse still improve over the baselines? If I compare to the results of the original PatchTST paper, I suspect that the improvement over PatchTST might be much smaller when ran with longer input length, at least for the long-term benchmark dataset.

2/ What would the full results look like for the TSFEL feature set? What improvement has TimeFuse over this feature set?

3/ Does TimeFuse improve over other AutoML/ensemble baselines?

4/ Would the TimeFuse ensemble method improve over stronger ensemble baselines (like forward selection)?

I would consider raising my score if the authors address these points.

**Relation To Broader Scientific Literature:**

As mentioned before previously, this work does not compare against other AutoML/ensemble methods like AutoGluon-TimeSeries and does not compare against strong ensemble baselines such as forward selection. Additionally, for the zero-shot results the paper does not mention and/or compare against recent zero-shot pretrained time series models (TimesFM, Chronos, or even TabPFN-v2). Thus, it is unclear whether the presented method improves over other AutoML packages or ensemble methods or the zero-shot results present improvements over pre-trained models.

**Theoretical Claims:**

There are no theoretical claims in this work.

---

> ### Author Rebuttal · Authors · 2025-03-31
>
> **Great thanks for your thoughtful review and constructive feedback! With our best efforts, we conducted numerous additional analyses to address your concerns. Full results: https://anonymous.4open.science/api/repo/ICML25Reb-034C/file/Results.pdf?v=b00206ad. Please understand that due to the 5000-character limit, we can only summarize the most critical findings in the rebuttal :)**
>
> # Q1: Experiment with long input length
>
> **Following your suggestion, we trained multiple base models and TimeFuse with longer input context lengths $L$ (both 336 and 512) on 7 long-term forecasting datasets.** Given limited time, we trained PatchTST and models newer than it except for TimeMixer, which encounters OOM on V100-32GB GPU when training with L=336 or higher.
>
> We report the averaged MSE for $L=96/336/512$ below (with 1st*/2nd best highlighted), please see full results for each dataset and metrics in **Table 4** in the link.
>
> |InputLen (L)|TimeFuse|PatchTST|TimeXer|PAttn|iTransformer|TimesNet|
> |-|-|-|-|-|-|-|
> |L=96|**0.257***|0.285|**0.269**|0.306|0.287|0.311|
> |L=336|**0.248***|0.266|0.270|**0.264**|0.288|0.324|
> |L=512|**0.245***|0.260|0.265|**0.260**|0.287|0.329|
>
> ## Key findings
> 1. **Across different $L$, TimeFuse consistently outperforms individual models by dynamically fusing their predictions.**
> 2. **Both PatchTST and PAttn benefit from longer input lengths.** At L=336/512, PAttn replaces TimeXer as the best model, with PatchTST showing comparable performance.
> 3. **TimeXer and iTransformer are insensitive to input length, while TimesNet degrades as input length increases.** Despite these mixed trends, TimeFuse effectively learns each model’s strengths across samples to produce more accurate predictions.
>
> Additionally, we want to highlight that we use TSLib for implementing all models with unified APIs and recommended hyperparameters. While there may be slight differences from the original PatchTST implementation, we confirm that **our PatchTST results align with (often better than) those reported in recent papers (e.g., TimeMixer)**.
>
>
> # Q2 & M1: Results and clarification on TSFEL feature set
>
> Please see **Table 6** in the full results for more results with the TSFEL feature set.
>
> **We want to clarify that our goal in proposing a new meta-feature set was NOT to outperform TSFEL (nor do we claim this as a core contribution)**, but rather to (i) avoid TSFEL's engineering issues and (ii) demonstrate that TimeFuse can integrate with various meta-feature sets and still perform well.
>
> More specifically, we propose the compact set alongside TSFEL for three main reasons:
>
> 1. **Unexpected NaN values in TSFEL**: We observed that TSFEL often outputs a large number of NaN values, e.g., 15,876 NaNs on the weather testset. **Table 5** in the full results shows NaN counts across datasets. TSFEL’s documentation offers no clear explanation or fix. In our implementation, we impute NaNs with the feature-wise mean.
> 2. **Lack of multivariate features in TSFEL**: TSFEL computes features based on single variable. We believe that inter-variable relationships are also important. Therefore, we constructed a smaller meta-feature set that explicitly includes such information.
> 3. **Demonstrating TimeFuse’s flexibility**: We see TimeFuse’s ability to work with different meta-feature sets as an advantage: with a well-engineered implementation, TSFEL and other existing sets (e.g., catch22) can all be seemlessly intergrated into TimeFuse. Users can also design their own feature sets based on domain needs or interpretability considerations.
>
> # Q3 & Q4 & M2: Comparison with AutoGluon, stronger ensemble algorithms, and foundation time-series model.
>
> Please see **Table 1** in the full results for a comprehensive comparison with the suggested baselines. **Due to space constraints, we kindly refer you to our response to Reviewer oty5 Q1 for detailed analysis.**
>
> **In short, TimeFuse outperforms the ensemble baselines with its test-time adaptive ensembling capability. AutoGluon and Chronos are limited by their univariate-centric design and short pretrain predict length, leading to especially poor performance on long-term or multivariate forecasting tasks.**
>
> # M3: Performance with raw features
>
> **Please see Table 6 in full results for the performance of using raw features.**
> Using either ours or TSFEL feature sets consistently outperforms the result with raw features by a significant margin.
>
>
> # D1: Discuss on the evaluation datasets
>
> **We greatly appreciate you pointing out the related works on the limitations of long-term time series benchmarks.** We have carefully read the literature and found them very insightful, relevant discussion will be included in the paper.
>
> **We want to thank you again for the thoughtful review. We’ve dedicated over 60 hours to get the new results and will include them to enhance the paper’s quality. We hope our responses address your concerns and are happy to discuss if you have any further questions!**

---

> > ### Comment · Reviewer_tYpH · 2025-04-03
> >
> > I would like to thank the authors for their rebuttal. I appreciate the much stronger baselines and updated settings to run those baselines that the authors selected to update their results. However, I agree with reviewer oty5 that the findings regarding AutoGluon/Chronos require some justification as they are inconsistent with other findings in both public leaderboards and literature.
> >
> > Nevertheless, I have increased my score.

---

> > > ### Author Response · Authors · 2025-04-03
> > >
> > > **Dear Reviewer tYpH,**
> > >
> > > **Thank you once again for your thoughtful review and encouraging feedback, it means a lot to us!**
> > >
> > > **Please also allow us to briefly clarify our findings regarding AutoGluon and Chronos:**
> > >
> > > 1. Both have only been evaluated on short-term univariate forecasting in prior work. To the best of our knowledge, **we are the first to evaluate them in a long-term forecasting setting (and it took us much effort)**, where we observed a prediction collapse issue with Chronos when the prediction length exceeds its pretrained horizon of 64.
> > >     > Note: In the original AutoGluon and Chronos papers, the prediction lengths vary across datasets without clear justification. On average, they are short: **18.07 (AutoGluon)** and **21.95 (Chronos)**, with some **as low as 4**, please see Table 8/3 in the respective papers at these links: [[AutoGluon](https://arxiv.org/pdf/2308.05566#page=18)] [[Chronos](https://arxiv.org/pdf/2403.07815#page=31)].
> > > 2. Neither method natively supports multivariate time series (MTS). Both AutoGluon and Chronos operate on a per-variable basis, requiring separate forecasting for each variable. This not only increases inference time but may also limit their performance compared to recent deep learning models designed for direct MTS forecasting.
> > >
> > > **We hope this further addresses your concerns and will include all new results and open-source the corresponding code for full transparency. Thank you again for your valuable support!**
> > > > We'd be truly grateful if you're open to reassessing in light of the new efforts we made to improve the paper :)
> > >
> > > **All the best wishes,**
> > > **Authors**

---

### Official Review · Reviewer_F3V1 · 2025-03-14

**Overall Recommendation:** 3

**Summary:**

This paper proposes TIMEFUSE, a framework designed to improve time-series forecasting accuracy by adaptively fusing the predictions of multiple (pre-)trained forecasting models. The core idea is to train a “fusor” model that predicts a suitable set of fusion weights based on a set of expert-designed meta-features extracted from each input time series. By leveraging statistical, temporal, and spectral descriptors to characterize each input sample, TIMEFUSE dynamically weights and combines the outputs of different base models, thereby taking advantages of each base model’s complementary strengths. Experimental evaluations on standard long-term (e.g., ETT, Electricity, Weather, Traffic) and short-term (e.g., PEMS traffic, EPF electricity price) forecasting benchmarks show improvements over state-of-the-art individual models. TIMEFUSE also empirically demonstrates its task-agnostic ability to zero-shot generalize to completely unseen datasets.

**Claims And Evidence:**

The main claims of the paper are (1). no single model can predict all data instances well, and (2). it is possible to train a simple model based on a set of expert-designed meta features to predict the combination weights for all model to get a adaptive predictions. Both claims seem intuitively reasonable and are backed up well with motivation examples and empirical results.

**Essential References Not Discussed:**

To my knowledge, this paper refers to a reasonably good number of references.

**Experimental Designs Or Analyses:**

The experiment designs, i.e., main results, ablation studies, sensitivity of the model zone size, seem reasonably complete and can empirically back up the advantage of the proposed method.

However, using multiple models instead of one single model will require more resource and time for both training and inference. The tradeoffs between the forecasting accuracy and the (a). training time and (2). inference time for different model zoo sizes should also be reported to the readers.

**Methods And Evaluation Criteria:**

The methods follows the main claims well.

The datasets, the experimental settings (look-back length and forecasting length) and the main evaluation metrics (MSE and MAE, etc.) follow the common practice in MTSF.

**Other Comments Or Suggestions:**

C1. Although TIMEFUSE demonstrates good average forecasting accuracy, it would be very useful if the authors can provide a per-instance breakdown of the best models, as in Figure 1. This can demonstrate how much TIMEFUSE is away from the ideal case, and relevant discussions about what might be missing can be very beneficial to understand the questions.

C2. I like Figure 4. Besides the weights, can you also combine the forecasting accuracies of each model in Figure 4, such that it would be very easy to see if TIMEFUSE assigns higher weights on more accurate models?

C3. I like Figure 6, too. Can the authors also combine the domain and properties of each dataset in this figure, to further enhance the interpretability of TIMEFUSE?

**Other Strengths And Weaknesses:**

S1. The paper is largely well-written and easy to follow.

Besides the experimental designs (forecasting accuracy vs. training and inference time under different model zoo sizes) to be clarified, some other weaknesses include:

W1. It seems the TSFEL feature set is pretty comprehensive and shows similar performance, why not just use it? What is the specific technical novelty of the TIMEFUSE meta-features compared with TSFEL?

W2. The training objective is not clear. Specifically, what is the optimal combination weight, and how to get it? Shall we choose the best single model, or should we choose a best weighted sum of the models, even though all models might be either overestimated and underestimated? How to break tie? Please clarify the details and justify the design choices.

**Questions For Authors:**

Q1. Ensemble learning has been a popular learning paradigm to enhance accuracy before the foundation model era. However, with the emergence of foundation models, a popular understanding is that large models after a certain scales can be much more powerful than ensembles of smaller models. Can the authors envision the possibility of large model for MTSF (actually there are already quite some explorations), and whether or how TIMEFUSE can still be useful with large MTSF models?

**Relation To Broader Scientific Literature:**

To my understanding, there are no clear relations to a potential broader scientific literature, besides MTSF and its applications.

**Theoretical Claims:**

To my understanding, there are no theoretical claims in this paper.

---

> ### Author Rebuttal · Authors · 2025-03-31
>
> **Thank you for the thoughtful review and constructive feedback! With our best efforts, we conducted numerous additional analyses to address your concerns. Full results: https://anonymous.4open.science/api/repo/ICML25Reb-034C/file/Results.pdf?v=b00206ad. Please understand that due to the 5000-character limit, we can only summarize the most critical insights in the rebuttal :)**
>
> # W1: Why not TSFEL
>
> We propose a compact meta-feature set alongside TSFEL for three main reasons:
>
> 1. **Unexpected NaN values in TSFEL**: We observed that TSFEL often outputs a large number of NaN values, e.g., 15,876 NaNs on the weather testset. **Table 5** in the full results shows NaN counts across datasets. These NaNs scattered across different samples and features. TSFEL’s documentation offers no clear explanation or fix. In our implementation, we impute NaNs with the feature-wise mean.
> 2. **Lack of multivariate features in TSFEL**: TSFEL computes features only on single variable. We believe that for multivariate time series, inter-variable relationships are important. Therefore, we constructed a smaller meta-feature set that explicitly includes such information.
> 3. **Demonstrating TimeFuse’s flexibility**: We see TimeFuse’s ability to work with different meta-feature sets as an advantage: with a well-engineered implementation, TSFEL and other existing sets (e.g., catch22) can all be seemlessly intergrated into TimeFuse. Users can also design their own feature sets based on domain needs or interpretability considerations.
>
> # W2: Training objective and other details
>
> - **Training objective:** Explicitly computing optimal model weights for each sample requires *solving a least squares problem for each sample*, which is computationally expensive and unnecessary. In practice, our training objective in Eq (1) can directly optimizes the fusor via backpropagation.
> - **How to ensemble:** Our goal is to learn the best weighted sum of base models. This has several advantages over selecting the best single model: (i) **no need to break ties** since we’re not picking just one model; (ii) **more flexible integration**: for example, if one model overestimates and another underestimates, their combination can outperform either model alone. In cases where all models over-/underestimate, our approach can fall back to single-model selection. In practice, such cases can be also mitigated by including more diverse models in the zoo.
>
> We will carefully revise the paper to better clarify these details.
>
> # C1: Sample-level breakdown analysis
>
> **Great suggestion! Please see Table 3 in the full results, where we report the sample-level win rate and average rank for all 13 base models and TimeFuse, using the same settings as Fig 1.**
>
> Overall, TimeFuse achieves a **46.11%** win rate with base models range **0.12%-9.89%**. For average rank, TimeFuse scores **2.37**, compared to **13.83–4.09** for base models, significantly outperforming any single model at the sample level. For cases where TimeFuse doesn’t outperform all base models, your point about “all models over/underestimating” is likely a cause. This might be addressed by adding a prefiltering step to select which models participate in the ensemble.
>
>
> # C2: Update Figure 4
>
> Please see **Figure 1** in the full results for an updated version, where we order the base models by performance. Generally, TimeFuse tends to assign higher weights to more accurate base models.
>
> # C3: Clarification on Figure 6
>
> Thank you for the suggestion. We note that Fig. 6 shows the fusor weights jointly trained across all long-term forecasting datasets. Its goal is to reveal the relationship between model weights and **input sample** properties, so the characteristics of a single dataset should not affect the results in Fig. 6.
>
> # Q1: On large MTSF models
>
> **We believe ensemble methods can complement large MTSF models rather than compete with them.** In fact, existing AutoML frameworks like AutoGluon already use pretrained MTSF models as base models in ensembles and achieve improved performance.
>
> Additionally, current large MTSF models still have notable limitations: for example, we found that Chronos (a pretrained MTSF model) struggles with long-term or multivariate forecasting due to its univariate-centric design. Please kindly refer to our response to Reviewer oty5 Q1 for more details.
>
>
> # W0: Accuracy-time tradeoff
>
> **Since TimeFuse's fusor has a very simple architecture, its computational cost is minimal compared to the base model.** As shown in the **Table 2** of full results, the batch inference time of TimeFuse is nearly identical to that of the base model, indicating negligible overhead. We will include a detailed discussion in the paper.
>
> **We want to thank you again for the thoughtful review. We’ve dedicated over 60 hours to get the new results and will include them to enhance the paper’s quality. We hope our responses address your concerns and are happy to discuss if you have any further questions!**

---

> > ### Comment · Reviewer_F3V1 · 2025-04-05
> >
> > Thank you for the response. My concerns are mostly addressed. I also agree with other reviewers' concerns on the evaluation setup. Under the expectation that these rebuttal discussions and experimental findings will be included in the revision for the final paper, I decided to increase my score.

---

> > > ### Author Response · Authors · 2025-04-05
> > >
> > > Dear Reviewer F3V1,
> > >
> > > Again, we thank you for your thoughtful review and encouraging feedback, it means a lot to us! We will incorporate all new results into the paper and open-source the corresponding code for full transparency. Thank you again for your support!
> > >
> > > All the best wishes,
> > > Authors

---

### Decision · Program_Chairs · 2025-05-01

**Decision:**

Accept (poster)

**Comment:**

This paper introduces TIMEFUSE, a framework designed to improve time series forecasting by dynamically fusing predictions from a diverse set of pre-trained base models ("model zoo"). The core idea is to extract a set of meta-features (statistical, temporal, spectral, multivariate) from each input time series sample and use these features to train a lightweight "fusor" model. This fusor predicts instance-specific optimal weights for combining the outputs of the base models, aiming to leverage the complementary strengths of different forecasters. Initial reviews acknowledged the novelty and soundness of the approach, its clarity, and good motivation. However, significant concerns were raised regarding the evaluation methodology, particularly the weakness of the initial ensemble baselines (simple mean/median), the lack of comparison to established AutoML frameworks (like AutoGluon) or stronger ensemble techniques (like Caruana selection), and the absence of comparison to recent large pre-trained time series models (like Chronos), especially concerning the zero-shot claims. Questions were also raised about the experimental setup (e.g., using a short, fixed input length for base models) and the evaluation protocol (reporting results primarily on datasets used for fusor training).

In response, the authors conducted a substantial rebuttal effort, including over 60 hours of additional experiments. They introduced comparisons against strong baselines including AutoGluon-TimeSeries, Chronos (zero-shot and fine-tuned), Caruana selection, portfolio optimization, and zero-shot ensembles across all 16 datasets. They also reran experiments with longer input lengths for base models. The new results showed TIMEFUSE consistently outperforming these enhanced baselines, with the authors providing justifications for the observed poorer performance of AutoGluon/Chronos on long-term multivariate tasks (univariate design, pretraining limitations). They clarified aspects of their methodology, including the training objectives, the rationale for their proposed meta-features versus TSFEL (NaN issues, lack of multivariate features in TSFEL) and raw features, and provided analysis on computational overhead (minimal). This extensive rebuttal was well-received: all three reviewers who had initially provided scores (ranging from Weak Reject to Weak Accept) explicitly raised their scores in light of the new evidence and clarifications.